
# Soil information and soil property maps for the Kurdistan region, Dohuk governorate (Iraq)

Mathias Bellat[1,2], Mjahid Zebari[3], Benjamin Glissmann[1,4], Tobias Rentschler[1,2,5], Paola Sconzo[6], Nafiseh Kakhani[1,2], Ruhollah Taghizadeh-Mehrjardi[1,2,7], Pegah Kohsravani[2,8], Bekas Brifkany[9], Peter Pfälzner[1,4], and Thomas Scholten[1,2,10]

[1]CRC 1070, University of Tübingen, Tübingen, 72070, Germany
[2]Department of Geosciences, Working group of Soil Science and Geomorphology, University of Tübingen, Tübingen, 72070, Germany
[3]Ludwig-Maximilians-Universität München, Müchen, 80634, Germany
[4]Institute for Ancient Near Eastern Studies (IANES), University of Tübingen, Tübingen, 72070, Germany
[5]Digital Humanities Center, University of Tübingen, 72074, Germany
[6]Department of Culture and Society, University of Palermo, Palermo, 90133, Italy
[7]Faculty of Agriculture and Natural resources, Ardakan University, Ardakan, Iran
[8]College of Agriculture, Shiraz University, Shiraz, Iran
[9]Dohuk Directorate of Antiquities and Heritages, Dohuk, Iraq
[10]DFG Cluster of Excellence "Machine Learning: New Perspectives for Science", University of Tübingen, Tübingen, 72076, Germany

**Correspondence:** Mathias Bellat (mathias.bellat@uni-tuebingen.de)

**Abstract.** We present the first detailed soil property maps at multiple depths for the northwestern autonomous Kurdistan region of Iraq (Dohuk). A total of 532 soil samples from 122 sites were collected at five depth increments (0-10, 10-30, 30-50, 50-70, and 70-100 cm), and their mid-infrared (MIR) spectra were measured. A subset of 108 samples, selected via Kennard–Stone sampling, was analysed in a laboratory on ten soil properties. A Cubist model was trained and used from these measured values
to predict all samples' soil properties from their MIR spectra. Digital soil mapping was conducted using various machine learning regression techniques (ensemble learning, linear classifier, nearest neighbour classifier, decision trees), trained on the predicted soil properties and using a total of 85 covariates at 25 m pixel resolution, resulting in 50 prediction maps in total. Results were compared with the *SoilGrids 2.0* product and a regional texture model. Soil depth was also mapped using a quantile random forest with 26 covariates. Our regional model outperformed global *SoilGrids 2.0* predictions in resolution
and accuracy, with texture RMSEs (sand: $\sum$RMSE = 9.35; silt: $\sum$RMSE = 6.8; clay: $\sum$RMSE = 10.28) comparable to local models. Quantile random forest achieved the best performance in 51 % of the models, and key predictors included Sentinel 2 SWIR, EVI, NDVI, and SAVI. Spatial patterns reflected the contrast between the flat areas of the Simele and Zakho plains, as opposed to the shallower and steeper Little Khabur Valley and anticline formations. Furthermore, the soil depth prediction model ($R^2$ = 0.57; RMSE = 2.59 cm$^{-0.5}$) showed strong correlation with slope and a similar pattern distribution with deeper
soils in the flat areas of the Simele and Zakho plains, while shallow soils are visible in the anticline and strongly erodible areas. Our comprehensive dataset (Bellat et al., 2024a, b, c, d, 2025) offers substantial insights for soil knowledge in the region, as well as for aridic and semi-aridic areas.



# 1 Introduction

Soils record chemical, physical and biological processes over extended temporal scales (Hillel and Hatfield, 2005; Schaetzl
and Anderson, 2005; Duchaufour et al., 2020). They are part of global exchanges (Bossio et al., 2020; Lal et al., 2021; Telo da
Gama, 2023) and exert significant influence local ecosystems (Adhikari and Hartemink, 2016; Scholten et al., 2017; Zeraat-
pisheh et al., 2022; Webber et al., 2023; Guan et al., 2024). Soil texture provides insights into soil stability, water retention,
carbon storage, and biomass production (Rabot et al., 2018), while pH regulates soil acidity and nutrient availability for plants
(Thomas, 1996; Neina, 2019). Organic carbon (Corg) reflects local organic production and functions as a major storage pit for
$CO_2$ at a global level (Trivedi et al., 2018; Bossio et al., 2020; Beillouin et al., 2023). Inorganic carbon — calculated as total
carbon ($C_t$) minus organic carbon — also plays a critical role in carbon sequestration in semi-arid zones (Zamanian et al., 2016;
Sharififar et al., 2023). Calcium carbonate ($CaCO_3$), abundant in calcareous soils of semi-arid climates, further influences both
acidity (Yu et al., 2023) and carbon dynamics (Umer et al., 2020; Dou et al., 2023). Additional key soil properties includes
total nitrogen ($N_t$), which influences plant growth (Crawford and Forde, 2002; Anas et al., 2020), and electrical conductivity
(EC), essential for assessing soil water content or capacity (Brevik et al., 2006), and soil salinity (Friedman, 2005), particularly
problematic in arid and semi-arid regions such as Iraq (Smith and Robertson, 1962; Christen and Saliem, 2013; Azeez and
Rahimi, 2017). Evaluating all of these properties and establishing a taxonomic classification of a soil gives information on
its ability to fit or not for agricultural purposes, but also to better understand the development of soils over time and under
changing climatic conditions.

In the Dohuk Governorate of north-western Kurdistan (**Figure 1**), exploratory mapping efforts (Buringh, 1957, 1960; Altaie,
1968; Altaie et al., 1969; Barzanji, 1973; Muhaimeed et al., 2014; Muhaimeed, 2020) identified the presence of semi-arid and
mountainous soils shaped by complex interactions between geomorphology, parent material and climate. The fluvial dynamics
of the Tigris River have been recognised as a major factor in landscape formation, influencing salinity, clay deposition, and
vertic properties through sedimentation and erosion (Buringh, 1960, pp. 51–54). However, critics (Wilkinson, 1990) have
suggested that vertic features and horizons might have been overestimated (Buringh, 1957, 1960; Altaie, 1968; Abdulrahman
et al., 2020). Gypsum is another critical factor in local soil development, either inherited from primary deposits such as alabaster
formations (Buringh, 1960, p.106), or formed secondarily through irrigation-induced precipitation and soil chemical processes
(Buringh, 1960, p.107). High gypsum concentrations are commonly found in areas south and south-west of the Zagros and
Taurus mountain chains (Smith and Robertson, 1962; Barzanji, 1973; Azeez and Rahimi, 2017), reflecting the influence of
regional hydrogeology, and aquitard structures (Buringh 1960, p.108; Azeez and Rahimi 2017). Favourable factors for soil
development have been poorly explored outside of the alluvial plain area (Altaie et al., 1969; Barzanji, 1973). While some
valley bottom soils may exhibit higher organic carbon content (Buringh 1960, p.78; Altaie 1968) soils in upland areas are
generally poorly developed due to severe erosion, leading to shallow, fragmented profiles (Muhaimeed et al., 2013) referred to
as "broken soils" (Buringh, 1957; Altaie, 1968).

Quantitative soil property data for the region remain scarce. The global *SoilGrids 2.0* product (Poggio et al., 2021) offers
coarse-resolution (250 m) predictions of key soil attributes. While adequate at a national scale in some regions (Varón-Ramírez



et al., 2022; Shi et al., 2025), its performance at finer scales is limited, particularly due to sparse calibration points in the Middle East and Iraq (Batjes et al., 2020; FAO and IIASA, 2023). At the local scale, only one recent study has attempted digital texture mapping (Yousif et al., 2023), but it covers a different area and does not account for the full range of soil-forming factors described in the *Scorpan* model (McBratney et al., 2003).

Previous classifications and soil descriptions in the region were mostly carried out at the national scale and do not reflect recent landscape changes (Forti et al., 2022), nor do they align with contemporary standards (WRB, 2006). Moreover, no high-resolution, spatially explicit dataset currently exists for the most important chemical and physical soil attributes. While the previous mappings only used limited observations windows, with modern digital soil mapping (DSM), the spatial distribution of soils and their characteristics can now be described and modelled with increasing accuracy (Behrens and Scholten, 2006; Taghizadeh-Mehrjardi et al., 2014). Therefore, we have developed a meso-scale (1:200,000; 25 m pixel) DSM of key properties in the Dohuk region, alongside an updated classification based on the WRB taxonomy (WRB, 2006). Soil sampling campaigns conducted between 2022 and 2023 enabled the creation of 10 soil property maps across five depth intervals and a soil depth model for the western part of Dohuk directorate (**Figure 1**). All data products follow the FAIR principles (Findable, Accessible, Interoperable, Reusable; Wilkinson et al. 2016) and were adapted to physical geography specificities (Bailo et al., 2020). These outputs are relevant for application in agriculture, geography, and ecology, especially as climate change exacerbates desertification in Iraq (Eltaif and Gharaibeh, 2022; Eltaif et al., 2024). The production of a high-resolution dataset and digital soil maps from recent field observations became an asset for depicting actual soil situations and exploring potential solutions.

## 2 Material and Methods

The workflow (**Figure 2**) followed a semi-standardised fully reproducible protocol (Malone et al., 2022). Using a cluster Latin hypercube sampling design (cLHS), 532 soil samples were collected from 122 sites (**Table 1**) and analysed *via* mid-infrared (MIR) spectroscopy. A representative subset of 108 samples, including legacy material from older surveys, underwent laboratory analysis for detailed physical, biological, and chemical characterisation. These samples were used to calibrate a Cubist regression model, with a raw and three transformed MIR spectra as predictor variables. The resulting predictions of soil properties were then integrated into a digital soil mapping framework (Lagacherie et al., 2006; Behrens and Scholten, 2006; Brevik et al., 2016; Malone et al., 2017; Hengl and Robert, 2019), following the *Scorpan* equation model (**Equation 2**; McBratney et al. 2003), and tested with six machine learning algorithms for each of five soil depth increments. Simultaneously, a soil depth map was developed using quantile random forest regression, incorporating remote sensing covariates and field observations. The digital soil maps integrate field observations and spatial predictions derived from a suite of remote sensing and spatial datasets. All datasets used are listed in **Appendix 1**, with further methodological detail provided in the supplementary material.





## 2.1 Study area

The data were collected from the Dohuk governorate in the Kurdistan region of Iraq, specifically from the Simele and Zakho districts (**Figure 3**) covering a total area of 2,280 km$^2$. The region is often referred to as Eastern Ḥābūr/Khabur (Pfälzner and Sconzo, 2016), though it is sometimes divided into two entities: the eastern Syrian al-Jazira/Ǧazīra for the western and southern part and the mountain chain of Ḥābūr/Khabur for the northern part (Abdulsalam and Schlaich, 1988).

### 2.1.1 Tectonic development and parent material

Our study area within the Dohuk governorate is located within the northwestern segment of the Zagros-fold thrust belt (ZFTB), a mountain belt that extends from southern Iran NW-ward to the Kurdistan Region of Iraq and SE Turkey. The ZFTB resulted from the ongoing convergence between the Arabian and Eurasian plates (Berberian, 1995; Agard et al., 2011; Mouthereau et al., 2012; Sembroni et al., 2024). The convergence started in the late Cretaceous with the subduction of the Neotethys oceanic crust beneath the Eurasia Plate and the obduction of the ophiolite sequences on Arabia's margin, followed by the subsequent continent-continent collision between the Arabian and Eurasian plates during the Oligocene-Early Miocene (Agard et al., 2011; Khadivi et al., 2012; Mouthereau et al., 2012; Koshnaw et al., 2017). Since the onset of continental deformation on the northeastern margin of the Arabian Plate (including the study area), it has propagated for 250–350 km (Blanc et al., 2003; Molinaro et al., 2005; Alavi, 2007; Agard et al., 2011; Mouthereau et al., 2012; Koshnaw et al., 2020; Zebari et al., 2020; Sembroni et al., 2024). Within the external part of the ZFTB, these zones include the Imbricated Zone, the High Folded Zone, and the Foothill (Low Folded) Zone (Berberian, 1995; Jassim and Goff, 2006; Fouad, 2012, 2014; Zebari et al., 2020).

The study area covers parts of the High Folded and Foothill zones (**Figure 4**), where structures are mainly trending in a nearly E-W direction (Forti et al., 2021; Doski and McClay, 2022). The Bekhair Anticline is the main structural and morphological feature in the area and plunges at the western end of our study area. It separates the Simele/Semel Plain, which stretches from the northern bank of the Mosul Dam Lake to the anticline, in the south, from the Zakho Plain and Little Khabur Valley to the north (Forti et al., 2021; Doski and McClay, 2022).

The exposed rocks in the area include sedimentary units ranging in age from the Upper Cretaceous to the Pliocene (Sissakian and Al-Jiburi, 2012, 2014; Doski and McClay, 2022). The Upper Cretaceous units consist of platform carbonates and siliciclastic rocks (Jassim and Goff, 2006; Aqrawi et al., 2010). The Paleocene-Eocene units consist mainly of marginal marine marls and shales that interfinger with rigid carbonate units, followed by red Eocene clays and carbonates. These Upper Cretaceous-Eocene units are exposed within the anticlinal structures in the area. The Oligocene units are missing in the area; thus, the Eocene carbonates underlie the Middle Miocene clays, evaporites, and limestones. The Upper Miocene–Pliocene units consist of fluvial sandy succession, clay, and conglomerate deposited in the Zagros foreland basin (Jassim and Goff, 2006; Aqrawi et al., 2010). The Miocene-Pliocene units are exposed within the synclines and low-elevation area to the north and south of Bekhair Anticline.

The Quaternary deposits cover three different environments of the study area (**Figure 4**). First, the flat area of the Simele Plain and north of the Zakho Plain (Türkiye) is covered by residual clayey soil material, coming from the erosion of the





Bekhair and Zagros anticlines. Second, along the riverbanks of the Tigris and Little Khabur rivers, sand and gravel-sized terrace deposits, as well as floodplain sediments of fine sand and clay, can be observed. Finally, Quaternary formations from alluvial fan sediments of clayey soil, combined with rock fragments coming from colluvial deposits, are visible in the foothills of the Bekhair and the shallow Little Khabur Valley. Sometimes, calcrete is also developed within the these Quaternary deposits (Sissakian and Al-Jiburi, 2012, 2014; Forti et al., 2021).

### 2.1.2 Climate and vegetation

The central part of the study area falls within a Csa (Hot-summer Mediterranean) agro-climatic zone, according to the Köppen Geiger classification (Köppen, 1936; Beck et al., 2018; Alwan et al., 2019). Annual precipitation ranges from 200 to 500 mm, with an average yearly temperature exceeding 16 °C (Fick and Hijmans, 2017; Salman et al., 2019; Najmaldin, 2023). Only the Little Khabur Valley, located north of the Bekhair anticline, experiences slightly cooler winters and receives higher rainfall,

typically between 500 and 800 mm per year (Fick and Hijmans, 2017; Alwan et al., 2019). South of the Bekhair anticline, the Simele Plain belongs to the Mesopotamian steppe floral complex, which supports a limited number of xerophytic shrubs and herbs, primarily *Artemisia herba-alba mesopotamica* often associated with *Aristida plumosa* (Guest and Al-Rawi 1966, pp.78-80; Zohary 1973, p.183). In contrast, the northern region, encompassing the Zakho Plain and Little Khabur Valley, falls within the Kurdo-Zagrosian climax zone, characterised by a denser xerophilous deciduous steppe forest, driven by its higher

elevation and more favourable climatic conditions. Dominant shrubs include *Anagyris foetida* or *Pistacia khinjuk* are associated with trees as *Quercus brantti*, or *Quercus boissieri*, which grows between 800 and 1,700 meters of altitude. Historical records mention the presence of pine forests (Zohary, 1973, pp. 183–190), though they are likely no longer extant. In both the foothills of the Mesopotamian Plain and the Kurdo-Zagrosian space, cultivated *Olivae europanis* can be sporadically observed.

### 2.1.3 Geomorphology and soils

In the southern part of our study area, the Tigris floodplain and its Quaternary alluvial deposits have largely disappeared due to the construction of the Mosul Dam Lake (Forti et al., 2022). What remains are sporadic surface exposures of conglomerates and marls (**Appendix 2**; Forti et al. 2021) and three to four terraces levels (Al-Dabbagh and Al-Naqib, 1991; Forti et al., 2021, 2022). North of the Tigris river, in the Simele plain, combined action of wind and irregular water action of *wādīs*, have led to the formation of gullies on this depositional glacis (Yacoub et al., 2012; Forti et al., 2021), shaping a badland landscape

(**Figure 5B**). The Bekhair anticline and its imbricated zone form a structurally homogenous ridge dominated by exposed limestone and sandstone formations (**Figure 4**; Forti et al. 2021), which are subject to lift-up process and tectonic action. The foothills on both sides of the ridge, however, are subject to wind, water erosion, and gravitational processes, resulting in extensive colluvial deposits (**Figure 5C**; Sissakian and Abdul Jab'bar 2014; Sissakian et al. 2015). In the area of the Tswoq anticline and the Little Khabur Valley, the landscape is dominated by sandstone and conglomerate. Soil surface erosional

process are less pronounced in the Little Khabur Valley region due to the protective effect of denser vegetation cover. The Zakho Plain, located within a synclinal structure (**Figure 4**), is a flat alluvial area, also less affected by erosion.



Soil mapping in the region was initially carried out in the 1950s and 1960s as part of the Iraq soil mapping project (Buringh, 1957; Altaie, 1968; Altaie et al., 1969). We adapted Buringh's classification to the WRB system (WRB, 2006), improved spatial detail using modern satellite imagery (Sentinel 2 ESA 2022; DEM ESA and Airbus 2022; and Bing Maps [https://www.bing.com/maps]), and completed the map with unrecorded Regosols and Fluvisols (**Figure 6**). The semi-arid climate, marked by sharp temperature variations and high subsurface $CaCO_3$ concentrations, has favoured the development of vertic and calcic features in many soil profiles (Abdulrahman et al., 2020). However, significant local variability exists. Soils adjacent to the Tigris River are typically calcic Vertisols, likely due to subsurface marl and conglomerate permeability. The Simele Plain's glacis deposits are dominated by cambic and gypsic Calcisols (**Figures 5 and 6**), with mediumly developed soil horizons, and a soil depth of 100 to 200 cm. North of the Simele plain, on the structural ridge and the steep slopes of the Bekhair anticline, soil development is minimal, due to active erosion, resulting in nudilithic Leptosols. On the northern side of the ridge, the Little Khabur Valley and its surroundings are dominated by poorly developed soil such as calcic Cambisols, Regosols and Leptosols, shaped by steep slopes and more erodible parent materials (conglomerate and sandstone), compared to the Simiele Plain. In contrast, the flat, irrigated Zakho alluvial plain, with higher precipitation, supports more developed soils, such as calcic isomeric Kastanozems (**Figure 6**). Finally, Fluvisols occur sporadically along the Tigris and Little Khabur floodplains and major $w\bar{a}d\bar{\imath}s$ channels riverbanks (**Figures 5 and 6**), identifiable by their ochric and/or umbric horizons.

## 2.2 Sampling campaign

The 2022 campaign primarily focused on the Simele Plain and the riverbank of the Tigris, while the 2023 mission was conducted in the Zakho district. Due to ongoing violence and conflict between the Kurdistan Workers' Party (PKK) and the Turkish government in the mountainous areas of Zakho (Ertan, 2022), the 2023 survey coverage was reduced for safety reasons. To increase the number of training samples for model calibration (see **2.4.1**), 16 additional sites and 29 samples were included from earlier 2017 - 2018 surveys (**Table 1**), and were based on purposive, non-randomised sampling design.

### 2.2.1 Conditioned Latin hypercube sampling

Sampling design plays a critical role in ensuring that selected locations reflect the spatial and environmental variability of the study area (Brus, 2022). We adopted a conditioned Latin hypercube sampling (cLHS) approach, a method particularly suited to digital soil mapping applications (Minasny and McBratney, 2006; Stumpf et al., 2016; Nketia et al., 2019; Wadoux and Brus, 2021). The cLHS method ensures that sampling points are distributed across the full range of values in selected environmental covariates by stratifying feature layers into equal intervals. Sampling was performed with R 4.4.0 (Team et al., 2024) using the clhs package (Roudier, 2012).

We selected six covariates (**Annexe 1**) which represent a broad range of parameters influencing soil variability. These included physical characteristics, underlying geomorphological formations (Forti et al., 2021), potential soil properties and erosion process.

The potential soil layer was constructed using spectral indexes (clay minerals, ferrous minerals, rock outcrop, carbonate) derived from climatic and satellite datasets (Copernicus, 2019; EROS, 2020). Erosion risk was modelled using the Revised



Universal Soil Loss Equation (RUSLE; Renard et al. 1991), incorporating five key factor: soil erodibility (K), soil coverage (C), topographic effect (LS), rainfall-runoff (R) and erosion control practices (P; Cossart et al. 2020; Thapa 2020; Abdi et al. 2023; Mehri et al. 2024).

### 2.2.2 Field measurements

At each site, samples were collected for the top 50 cm using a 3.5 cm ⌀auger and from depths up to 100 cm with a 2 cm
⌀auger. The different layers' depths were measured, and colour was determined according to the Munsell soil colour chart. Samples were collected at five depth increments: 0 - 10 cm, 10 - 30 cm, 30 - 50 cm, 50 - 70 cm and 70 - 100 cm. Bulk density was calculated for the topsoil using a 5.3 cm ⌀ring (Blake and Hartge, 1986). All samples were air-dried at 40 °C for 48 hours before sieving at 2 mm for subsequent analysis.

### 2.3 Laboratory analysis

### 190 2.3.1 Mid-infrared spectroscopy

Mid-infrared spectroscopy to measure physical and chemical soil properties has significantly evolved over the past decades (Ng et al., 2022a) and offers reliable results while saving time and resources (Stenberg et al., 2010; Viscarra Rossel et al., 2022). The soils samples were ground under 1 μm with a *Pulverisette 5/4, classic line* (Fritsh, Idar-Oberstein, Germany) before being pressed into a tablet, mixing 1 - 1.3 mg of soil and 250 mg of potassium bromide (KBr). The spectra were analysed with a
195 *Vertex 80v* (Bruker OPTIK GmbH, Germany), with a 4 cm$^{-1}$ resolution, on the 375 - 4,500 cm$^{-1}$ interval.

The spectra were imported into `R 4.4.0` and analysed using the `prospectr` (Stevens and Ramirez-Lopez, 2014) and `simplerspec` (Bauman, 2024) packages. To reduce noise interference, we decided to remove the measurements between 375 - 499 cm$^{-1}$ and 2451 - 2500 cm$^{-1}$ intervals and spectra value higher than 2 and lower than - 2 (Curran et al., 1996; Ng et al., 2018). The soil spectra were enhanced by applying three spectral transformations (Ng et al., 2018; Wadoux et al., 2021;
Ludwig et al., 2023), Savitzky-Golay with a polynomial order of 2 and a window size of 11 (SG 2.11), a moving average of 11 and standard normal variate transformation on the SG transformed spectra (SNV-SG). A total of 108 samples were selected for laboratory measurements **(Table 1)** using Kennard-Stone sampling (Kennard and Stone, 1969), ensuring a high diversity and variability of individuals, based on their spectral data (Ramirez-Lopez et al., 2014).

### 2.3.2 Soil properties

Seven properties were measured: pH, CaCO$_3$, N$_t$, C$_t$, Corg, EC and texture (**Table 2**). The pH was measured using a potassium chloride (KCl) solution, with a *ProfiLine pH 3310* and a *WTW SenTix 81 pH* electrode (Fisher Scientific, Strasbourg, France). Carbonate calcium (CaCO$_3$) content was determined as a percentage using a calcimeter *08.33* (Royal Eijkelkamp, Giesbeek, Netherlands). Total nitrogen (N$_t$), total carbon (C$_t$) and total organic carbon (Corg) were quantified as percentages with a CNS analyser, *Vario EL III* (Elementar, Hanau, Germany). The electro-conductivity (EC) was measured in micro-siemens per
centimeter (μS/cm) using a *Cond 330i/340i* (WTW, Weilheim in Oberbayern, Germany). Texture property was determined as a





percentage and measured through wet sieving for sand fraction and a *SediGraph III* for finer fractions (Micromeritics, Norcross, USA). Additionally, we estimated the mean weight diameter in mm (MWD, **Equation 1**) based on the texture results.

## 2.4 Models and pre-process

### 2.4.1 Spectra prediction

The Cubist model is a regression-based machine learning algorithm that extends the ideas of decision trees by combining rule-based predictive models with linear models at the leaves, enhancing both interpretability and predictive accuracy (Quinlan, 1992). This model excels at handling both continuous and categorical data, providing robust predictions even in the presence of complex interactions and non-linear relationships (Kuhn and Quinlan, 2024). Cubist's strength lies in its ability to partition the data space and fit separate linear models to each segment, making it particularly effective for problems with distinct patterns

or heteroscedasticity (Wang and Witten, 1996). This model has been applied in a variety of studies for soil property prediction from spectral predictors, such as (Viscarra Rossel et al., 2016; Padarian et al., 2020; Behrens et al., 2022). We tested a Cubist regression model on four spectral datasets in a `Python` (Foundation, 2022) environment using the `Cubist` library (Aselin, 2024).

### 2.4.2 Digital soil properties mapping

We based our soil property model on the soil formation factors of the *Scorpan* equation (**Equation 2**) developed by McBradney et al. (2003). We included 85 covariates (**Table 3** and **Appendix 2**). The remote sensing variables were accessed through Google Earth Engine (https://earthengine.google.com) and the different index computed in `R` with the `terra` (Hijmans et al., 2025) and `raster` package (Hijmans, 2010). The terrain variables were computed on *SAGA GIS 9.3.1* (Conrad et al., 2015) based on a filled and filtered DEM from GLO-30 ESA and Airbus (2022). All the computation part was realised under `R`

`4.4.0` environment (Team et al., 2024). As an input, we included 122 samples for the 0 - 10 cm depth, 112 for the 10 - 30 cm increment, 108 for the 30 - 50 cm depth, 98 for the 50 - 70 cm increment and 92 for the 70 - 100 cm depth. We divided the mapping of each variable for each soil depth increment, resulting in 50 models and maps in total. We performed a standardisation of the predicted values of the texture on 100 % with `TT.normalise.sum` function (Moeys et al., 2024) and a additive-log ratio transformation (Aitchison, 1986) with the `alr` function (Tsagris et al., 2025). This transformation

preserved the spatial information of the prediction with a repartition close to a normal distribution Liu et al. (2022). Digital soil mapping have adapted this additive-log ratio on the texture with success, alr_sand = $ln(\frac{sand}{clay})$ and alr_silt = $ln(\frac{silt}{clay})$ 2021; 2022. We also scaled the covariates with `preProcess` function and the `"range"` method from the `caret` package (Kuhn, 2019).

During the pre-processing, we performed a feature selection with the `Boruta` package (Kursa and Rudnicki, 2010). Using

a random forest-based model, `Boruta` validated or rejected the selection of variables regarding their influence on the inputs (**Appendix 3**). This method improves model accuracy and reduces overfitting results (Kursa and Rudnicki, 2010), and its efficiency has been proven for digital soil mapping (Taghizadeh-Mehrjardi et al., 2020; Suleymanov et al., 2024; Bouslihim





et al., 2024). We also performed a recursive feature elimination (RFE; Guyon et al. 2002) on the covariates with the `caret` package (Kuhn, 2019). However, the selection of covariates was not restrictive enough compared to the `Boruta` selection and was not retained.

Data were split into 80 % training and 20 % testing for each soil depth before being processed using a 10-fold cross-validation repeated three times. We tested six different models based on the state-of-the-art the art (Taghizadeh-Mehrjardi et al., 2016, 2020; Varón-Ramírez et al., 2022; Zolfaghari Nia et al., 2022; Khosravani et al., 2023; Shi et al., 2025): classification and regression tree (CART), $k$-nearest neighbours (KNN), support vector machine with a radial basis function kernel (SVMr), Cubist model, quantile regression forest model (QRF), and an ensemble model. All these models were implemented with the `caret` (Kuhn, 2019), `quantregForest` (Meinshausen and Michel, 2020), `Cubist` (Kuhn and Quinlan, 2024) and `caretEnsemble` (Deane-Mayer, 2024) packages.

Regression trees based on the CART model (Breiman et al., 2017) use a tree-based structure, splitting the data into different nodes. In the end, the model evaluates the leaves and selects those with the best performance. The key distinction of regression trees lies in their prediction of continuous values rather than classes at the terminal nodes, unlike classification trees. This simple and comprehensive model has been widely used for digital soil mapping (Taghizadeh-Mehrjardi et al., 2016; Zeraatpisheh et al., 2022; Zolfaghari Nia et al., 2022). The KNN algorithm is a non-parametric method that estimates the new values based on the closest input in an Euclidian distance. This model is widely used in digital soil mapping for its relative "simple" principle and the limited number of hyperparameters, which reduce its computing time. SVMr is a basic support vector machine using a linear regression (Drucker et al., 1996) to which a kernalisation of the data has been applied. The data are transformed into a high-dimensionality feature space, and a linear regression hyperplane is performed before the data are re-transformed into non-linear space. This radial basis function kernel is adapted for non-linear problems and complex data such as soil mapping (Taghizadeh-Mehrjardi et al., 2020; Pereira et al., 2022; Kaya et al., 2022). The cubist abilities as a regression model have been depicted above (cf. **2.4.1**). Its use in digital soil mapping has shown prominent results for many years (Taghizadeh-Mehrjardi et al. 2016; Malone et al. 2017, p.133; Hengl and Robert 2019, p.238). Based on random forests, the quantile regression forest model (Breiman, 2001), tracks each sample's value at each node, providing a conditional response distribution. This model is specially fitted to evaluate the accuracy with a prediction interval (Vaysse and Lagacherie, 2017; Varón-Ramírez et al., 2022). Finally, we produce an ensemble model based on the five above results, which ponders each model prediction to select the best one depending on the conditions. This "meta-model" was developed based on stacked regression computed with a random forest algorithm using the `caretStack` function (Deane-Mayer, 2024). This stacked regression approached is developed in two digital soil mapping packages `machisplin` (Brown, 2023) and `landmap` (Hengl, 2022). This method has proven efficient for digital soil mapping, especially in large-scale, regional or national, contexts (Varón-Ramírez et al., 2022).

### 2.4.3 Soil depth mapping

To predict soil depth, we developed a prediction model using remote sensing data and ground-truth control points, collected during surveys to estimate soil depth. The soil depth was measured from 0 to 100 cm on the 122 sampling sites; we added 25 zero values from remote sensing imagery observation on bare rock points. Soil depth is mainly determined by climate, terrain,





parent material, vegetation, and land uses (Zhang et al., 2021; Liu et al., 2022). Consequently, we used 25 environmental covariates to predict the soil depth (**Appendix 2**). Original soil depth data were first square root-transformed, and covariates were also scaled using the same method as for the soil properties mapping. Input data were split into 80 % training and 20 %

testing, using 10-fold cross-validation repeated three times. A quantile regression forest model (Meinshausen and View Profile, 2006) was chosen and implemented in the `R 4.4.0` environment (Team et al., 2024) using the `caret` (Kuhn, 2019) and `raster` (Hijmans, 2010) packages. As described above (cf. **2.4.2**), the QRF model is fitted for digital soil mapping. Contrary to the soil properties mapping where only the `mtry` parameter was tuned, for soil depth prediction, we also customised the minimum node size parameter `nodesize` and the number of trees was set as default at 500(Liu et al., 2022).

$$MWD = Xi * Wi/100 \qquad (1) \qquad\qquad S_a = f(s,c,o,r,p,a,n) \qquad (2)$$

### 2.4.4 Evaluation criteria

To evaluate model efficiency and precision, we used common metrics in spectroscopy prediction (Bellon-Maurel et al., 2010; Williams et al., 2017) and DSM (Lilburne et al., 2024). The most widely used is the root mean square error (RMSE, **Equation 3**), which measures the prediction ability of a model. The coefficient of determination, also called rsquared ($R^2$, **Equation 4**),

indicates the proportion of dispersion of the predicted vs. observed values. The means square error (MSE, **Equation 5**) assesses the risk of the estimator and was used only for spectra prediction, while the mean absolute error (MAE, **Equation 6**) calculates prediction accuracy. The concordance correlation coefficient evaluates the reproducibility of the model (CCC, **Equation 7**, Lin 1989). We also computed the ratio of performance to InterQuartile distance (RPIQ, **Equation 8**) for the spectra prediction, informing of the model validity. Finally, for the QRF models, we used the prediction interval coverage probability to evaluate

the corresponding prediction within an interval, here set at 90 % (PICP, **Equation 9**, Shrestha and Solomatine 2006; Malone et al. 2017, p.176.Vaysse and Lagacherie 2017).

$$RMSE = \sqrt{\frac{\sum_{i=1}^{n}(y_i - \hat{y}_i)^2}{N}} \qquad (3) \qquad\qquad R^2 = 1 - \frac{\sum_{i=1}^{n}(y_i - \hat{y}_i)^2}{\sum_{i=1}^{n}(y_i - \bar{y}_i)^2} \qquad (4)$$

$$MSE = \frac{1}{n}\sum_{i=1}^{n}(Y_i - \hat{Y}_i)^2 \qquad (5) \qquad\qquad MAE = \frac{1}{n}\sum_{i=1}^{n}|Y_i - \hat{Y}_i| \qquad (6)$$

$$CCC = \frac{2S_{XY}}{S_X^2 + S_Y^2 + (\bar{X} - \bar{Y})^2} \qquad (7) \qquad\qquad RPIQ = (Q3 - Q1)/RMSE \qquad (8)$$

$$PICP = \frac{1}{v} \ count \ j \quad j = PL_j^L \le t_j \le PL_j^U \qquad (9)$$



## 3 Results

### 3.1 Soil properties spectra prediction

The SNV-SG transformed spectra provided the best performance for six out of the ten soil properties, while SG-SNV and raw spectra were optimal for two properties each (**Table 4**). To evaluate prediction reliability, we applied the classification system proposed by Ng et al. 2022b, which categorises model performance from A (very reliable) to D (poor reliability), based on metric scores. In our results, $CaCO_3$, $C_t$, sand, and clay predictions fall within categories A and B, indicating high accuracy ($R^2$ = 0.72 - 0.91) and concordance (CCC > 0.80). Category C predictions include pH, $N_t$, Corg, and MWD, with moderate reliability. These models yielded $R^2$ values between 0.44 and 0.83, CCC values ranging from 0.60 to 0.89, and RPIQ scores from 1.15 to 1.71. In contrast, EC and silt content fall into category D ($R^2$ = 0.29 - 0.38; CCC = 0.49 - 0.56; RPIQ < 1).

The predicted soil texture predominantly corresponds to silty-clay and silty-clay-loam classes (**Figure 7**), with average silt and clay contents of 45.95 % and 38.39 %, respectively (**Table 2**). The distribution of most predicted properties closely mirrors that of observed values, except for EC, where outliers were removed during pre-processing (see supplementary material).

### 3.2 Digital soil properties mapping

QRF outperformed other models in 23 cases (51 %; **Table 5**), followed by the ensemble model (15 %), Cubist (13 %), and altogether KNN, SVMr, and CART performed better only in nine cases (20 %). Model performance was generally lower in the 0 - 10 cm depth increment (**Table 5**). However, exceptions were observed for Corg and alr_silt, which yielded higher CCC values (0.55 and 0.48), $R^2$ (0.32 and 0.49), and lower RMSEs (0.38 and 0.20) compared to deeper layers.

Features selection using `Boruta` significantly reduced the number of covariates by 77 - 93 % (**Table 6**). The most influential factors were Sentinel 2 EVI, NDVI, SAVI indexes, SWIR bands and potential evapotranspiration. The channel network base level variable was particularly important for predicting $N_t$ and Corg. Sentinel 2 products consistently outperformed those from Landsat 8. These covariates align with the *s* (soil), *o* (organisms), and *r* (relief) components of the *Scorpan* model.

The spatial distribution of soil properties shows a clear division between the southern and western parts of the region, Simele and Zakho plains, Tigris riverbanks, and the northern and eastern zones, Bekhair anticline and Little Khabur Valley, (**Figures 9, 10, 11, 12 and 13**).

While pH remains relatively stable with depth and shows higher values in the anticline and valley areas, except at 50–70 cm depth, where spatial variability increases (**Figure 12**), the $CaCO_3$ exhibits a uniform distribution across all depths, with isolated high values along the southern foothills of the anticline. $N_t$ shows elevated concentrations in the topsoil (0 - 30 cm), especially in the anticline and north-western zone (**Figures 9 and 10**). $C_t$ remains consistent across the first four depth intervals, peaking near the anticline, while the 70 - 100 cm layer reveals additional hotspots in the Simele Plain (**Figure 13**). The Corg varies with depth, with maxima shifting from the anticline (0 - 10 cm) to the Little Khabur Valley (50 - 70 cm), and mixed high values in other intervals. EC is highest in the Simele and Zakho plains, with moderate peaks in the Little Khabur Valley; at 70 - 100 cm, the highest values concentrate in central Simele and Zakho (**Figure 13**). Finally, texture distributions show higher sand content



in surface layers (0 - 10 cm) and more silt in the 50 - 70 cm increment. Sand is more prevalent in the anticline and valley areas, while silt and clay dominate in the plains. MWD follows the texture trends, peaking where sand content is higher.

## 3.3  Soil depth mapping

The soil depth model, $R^2$ explains approximately 57 % of the variations of soil depth, and the RMSE indicates a variability of the predicted value of 2.59 $cm^{-0.5}$. The CCC of 0.74 and MAE of 1.54 suggest a good agreement between the predicted and observed values. The model's uncertainty was well-calibrated, as reflected by a PICP of 90.87 % (**Table 5**). Predictors derived from the digital elevation model (DEM), were the most influential covariates, with four of the five highest-ranking variables
accounting for 30 % of the explained variance (**Table 6**).

## 4  Discussion

### 4.1  Spatial interpretations

The contrasting distribution of soil properties across the study area (**Figures 9, 10, 11, 12 and 13**) can be attributed mostly to landscape differences. The Simele and Zakho plains, together with the Tigris alluvial valley, largely comprise flat areas
along rivers and depressions, which are mainly characterized by sedimentation processes, for example the deposition of flood sediments on river bakns and terraces or the filling of depressions with erosion material. In contrast, the little Kabur Valley experiences stronger erosional processes with the formation of rills and gullies, and the Bekhair and Zagros anticlines are subject to uplift at the geological timescale.

The spatial correlation between pH and EC is particularly evident. Neutral pH values and elevated EC are mainly associ-
ated with Kastanozems in the Zakho Plain and calcareous Calcisols or Vertisols in the Simele Plain. Organic carbon and $N_t$ concentrations are higher in the Little Khabur Valley and mountainous areas, likely linked to denser vegetation cover (*Quercus brantii*, *Quercus boissieri*; Zohary 1973, pp. 183–190) and a cooler climate regime. $CaCO_3$ content appears to reflect the lithological composition of the Simele Plain, particularly the carbonate-rich sandstone Injana Formation. Consequently, total carbon ($C_t$), closely follows $CaCO_3$ distribution, with the majority of carbon storage deriving from inorganic forms in deeper
horizons (Moharana et al., 2021). Textural patterns reveal finer soil fractions (clay and silt) prevailing in the flat plains and the southern foothills of the Bekhair anticline. In contrast, coarse-textured soils with higher sand content are found in more eroded and badlands areas, such as the top of the anticline and the Little Khabur Valley.

### 4.2  Soil depth distribution and prediction uncertainty

Two distinct patterns emerged in the spatial distribution of soil depth (**Figure 14**). Shallow soils are prevalent in foothill and
mountainous regions, as expected (Patton et al., 2018), but are also common in badlands and along active *wādīs* channels, and riverbanks of the Tigris and Little Khabur.





In contrast, deeper soils were mapped in the Zakho Plain and on the plateaus of the Little Khabur Valley. These patterns may be explained by flat topography, active depositional process, and in the case of the plateaus, by denser vegetation zones typical of the Kurdo-Zagrosian climax formation.

The soil depth uncertainty map confirms the model's robustness in predicting both shallow and deep profiles. The highest uncertainty was found near major $w\bar{a}d\bar{\imath}s$, badlands, and along the foothills—areas with greater geomorphological variability and microrelief. The top-ranked covariates, mainly derived from DEM, confirm the well-established relationship between topography and soil depth (Patton et al., 2018; Yan et al., 2018; Liu et al., 2022).

### 4.3  *SoilGrid 2.0* product comparison

To assess model performance, we compared our results with the global *SoilGrids 2.0* product (Poggio et al., 2021), focusing on pH, Corg, $N_t$, and texture attributes, for three generalised depth intervals (0 - 30 cm, 30 - 60/70 cm, and 60/70 - 100 cm). Prior to comparison, values outside standard deviations in *SoilGrids 2.0* were replaced by median values (**Table 7**).

Our models predicted higher values of Corg and $N_t$, with respective increases of 900 % and 357 % over those from *SoilGrids 2.0*. Predicted sand and silt values were also slightly higher (by 17 % and 8 %), while clay and pH were slightly lower (by 17

375 % and 4 %).

Bivariate comparison maps (**Figure 8**) indicate that our model yields higher values of pH, $N_t$, Corg, and sand in the Simele Plain, while *SoilGrids 2.0* shows higher values for the same properties in the Little Khabur Valley. The spatial patterns of silt and clay are inversely distributed. Areas of similarity between the models include $N_t$ and Corg in the north-western part of the Little Khabur Valley, pH in the Bekhair anticline and eastern sector, and texture components in the upper Simele Plain.

Due to the use of different algorithms for each depth increment, a direct ensemble model comparison with *SoilGrids 2.0* (Varón-Ramírez et al., 2022) was not feasible.

### 4.4  Data quality, limitations, and future applications

The laboratory-measured soil properties and their corresponding FTIR spectra constitute a valuable and reusable dataset, crucial for improving predictive model performance over time (Viscarra Rossel et al., 2016; Safanelli et al., 2025). Regarding

FTIR-based predictions for $CaCO_3$, $C_t$, sand, and clay, they were classified as highly reliable (categories A–B; Ng et al. 2022b), making them suitable for research and applied uses. The pH, $N_t$, Corg, and MWD predictions (category C) should be limited to exploratory or screening applications. As for silt and EC they fall under category D and should be interpreted with caution—consistent. However, these two properties are especially known for having low-accuracy results in predictive models based on FTIR spectra (Hobley and Prater, 2019; Ng et al., 2022b).

The updated soil classification map (**Figure 6**) must be interpreted with care, specially at micro-scale (<1:50,000). First, arround 90 % of field observations were based on auger sampling rather than full-profile descriptions. Second, the 100 cm depth limit may omit deep horizons, although research from Abdulrahman et al. 2020 suggest that such horizons are uncommon in this region. Third, the Tigris right bank was mapped using remote sensing only, which may introduce higher uncertainty.





But despite these limitations, the current product offers improved detail over earlier maps (Buringh, 1957; Altaie, 1968), and
adheres to modern WRB standards (WRB, 2006).

Direct comparison of our model's accuracy with global datasets such as *SoilGrids 2.0* is inherently difficult due to differences
in spatial resolution and input data. While *SoilGrids 2.0* aims to provide consistent global coverage, our maps, with a resolution
of 25 m, offer substantial improvements for regional or local applications. Notably, the density of training samples used in our
study (53.5 per 1,000 km$^2$) greatly exceeds that of the *WoSIS* dataset used for *SoilGrids 2.0*, which reports a density of only
0.032 per 1,000 km$^2$ and includes no samples from the Kurdistan Region (Batjes et al., 2020). Our density of samples is close
to defined standard for regional studies maps (Hazelton and Murphy, 2016, p.5). Furthermore, the use of a conditioned Latin
hypercube sampling strategy further enhances spatial representativeness compared to legacy sampling methods (Brus, 2019;
Ma et al., 2020; Wadoux and Brus, 2021).

One limitation lies in the harmonisation of our soil depth intervals with other models (Arrouays et al., 2014; Poggio et al.,
2021; Varón-Ramírez et al., 2022; Shi et al., 2025). A second limitation concerns our limited depth observation window of 100
cm, whereas global products such as *SoilGrids 2.0* and the Chinese Soil Atlas extend to 200 cm (Shi et al., 2025).

Finally, compared to the local prediction model by Yousif et al. 2023, our RMSE values for sand and silt (at 0 - 10 cm depth)
are comparable (sand = 9.14; silt = 7.18). However, their clay predictions are more accurate (RMSE = 3.70), and overall model
R$^2$ is higher (sand = 0.91; silt = 0.85; clay = 0.90). Yet, their model is limited to topsoil (0 - 30 cm) and focuses only on "soil"
areas (202 km$^2$), based on LU/C classification, while our model covers a broader range of landscapes.

## 5   Data and repository organisation

The supplementary files of this paper contain all additional information and original product divided into eight folders:

1. RUSLE part contains all factor of the RUSLE model map and the map itself.

2. The cLHS part includes the `R` code and the soil profile points produced for 2022 and 2023 campaigns.

3. The field part contains all the photographs of sampling sites and the raw observations made during the campaigns,
   including the soil classification map in `.gpkc` format.

4. FITR element includes all the raw spectra in the `.dpt` format and the `R` code used to compile and filter these spectra.

5. Laboratory folder has only one item, the `.csv` of all the laboratory measurements detailed.

6. The spectra prediction folder includes the `Python` codes used to predict the soil properties based on the FTIR spectra
with a Cubist model, and the predictions results and metrics.

7. DSM folder contains all the codes and exportations from the digital soil mapping made under `R` including the raw
   prediction and uncertainty maps.

8. The Soil depth folder is similar to the DSM folder, only with soil depth values.



## 6 Code and data availability

The final maps products are available at https://doi.org/10.1594/PANGAEA.973764 (Bellat et al., 2024a) in both Network
Common Data Form 4 (NetCDF4) and GeoTIFF (GTiff) formats. Profiles depth measurement (https://doi.org/10.1594/PANGAEA.973714,
Bellat et al. 2024d), laboratory measurement (https://doi.org/10.1594/PANGAEA.973701, Bellat et al. 2024c) and MIR spec-
tra and its predictions (https://doi.org/10.1594/PANGAEA.973700, Bellat et al. 2024b) are also accessible online. All the
supplementary files and raw material is available at doi.org/10.57754/FDAT.e2k10-sf012, Bellat et al. 2025, and the interactive
material visible at https://mathias-bellat.github.io/DSM-Kurdistan/. Code is available in the supplementary material but also at
the GitHub deposit https://github.com/mathias-bellat/DSM-Kurdistan.git. Finally, we developed an online version of the pre-
diction maps of the soil properties, adapted for colourblind persons accessible at https://mathias-bellat.shinyapps.io/Northern-
Kurdistan-map.

## 7 Conclusions

We developed a full workflow for digital soil mapping at a regional scale in the Dohuk Directorate of the Kurdistan region of
Iraq. From the cLHS strategy, we selected 122 soil profiles at five depth increments (0 - 10 - 30 - 50 - 70 - 100 cm) and analysed
ten of their physical-biological or chemical properties based on their MIR spectra. Using these 531 samples, we predicted 50
maps using different machine-learning models at a resolution of 25 m per pixel. Additionally, we also produced a soil depth
map based on the QRF model and a detailed soil classes map at a 1:200,000 scale.

Results of the soil property models were compared to *SoilGrid 2.0* product and locally produced maps. Our models present
more adapted results for local interpretation than world products and are similar in accuracy to other local models, although
previous local models are limited in the number of soil depth increments, area size and properties analysed. As for the soil
classes map it fits modern WRB taxonomy standards and benefits from more observation and shallower resolution than previous
studies.

Spatial distribution of the soil properties and soil depth is globally divided into two distinct landscape units. One group
gathered the flatter area of the Tigris alluvial plain, and the Zakho and Simele plains had finer-grain-size particles (silt and
clay), and higher concentrations of calcium carbonate were observed. Conversely, the second group comprises erodible and
tectonically active Bekhair and Zagros anticlines, and Little Khabur Valley, presenting a coarser texture grain size and more
organic carbon due to their heavier vegetal coverage and lower anthropic impact.

## 8 Author contributions

P.F. and T.S. secured the funding; M.B., B.G., R.T.M., P.F. and T.S. conceived the study protocol; B.B. and P.F. assured the
security protocol during the sampling campaigns; M.B., B.G., T.R. and P.S. conducted the sampling campaigns; MB, TR, NK,
RTM and PK performed the analysis; M.B. interpreted the results; M.B., T.R., N.K., R.T.M. curated the data; M.B., M.Z., N.K.
and T.S. wrote the original draft of the manuscript; all authors contributed in the reviewing of the original draft manuscript.

## 9 Acknowledgments

The authors would like to thank all data contributors, especially the Directorate of Antiquities of Dohuk and its personnel, who assisted us during the sampling campaigns and without whom this project would not have been possible (Mohammed, Azad, Ali, and Walad) but also the students who helped on the field (Mathis, Marie-Amandine and Tom). We would also like to thank the *Humanum* consortium and Tara Beuzen-Waller who have left us access to their computing capacities.

## 10 Competing interests

The contact author has declared that none of the authors has any competing or conflict of interests.

## 11 Financial support

This work has received funding from the Deutsche Forshungemeischaft (DFG) Collaborative Research Center (CRC) 1070 "ResourceCultures", grant agreement n°215859406

## 12 Computing capacities

The sampling design and Cubist predictive model or spectra transformation were computed with: OS = *Windows 11*, CPU = Intel I7-9750H 2.60GHz, RAM = 32 GB 2667 MHZ DDR4 (1 - 2h process). For the prediction maps models were computed on the *Humanum* cloud infrastructure: OS = *CentOS 8 (arch x86_64)*, CPU = 2 * AMD EPYC 7542 32-Core Processor 2.9 MHz(64 cores), RAM = 16 * 64 GB 3200 MHz DDR4 (1024 GB), GPU = 2 * Nvidia A100 40G, (6h process per map, total 24h).

## 13 Carbon footprint

We calculated the estimated carbon footprint for all the different processes and steps needed for this publication and the analysis related to it:

– Field missions: 2,960 kg $CO_2$.

– Communications: 28 kg $CO_2$.

– Laboratory analysis: 77 kg $CO_2$.

– Computing capacities: 2.8 kg $CO_2$.

– Online research: 0.264 kg $CO_2$.



## 14 Artificial intelligence statement

Generative AI chatbot *GPT-4-turbo* and *Claude Sonnet4* were used to generate part of the R code and improve English writing.

## 15 Sample availability

All soil samples are conserved at the University of Tübingen in the Laboratory for Soil Science and Geoecology at Rümelinstraße 19 – 23, 72070, Tübingen, 72070, Germany. They can be consulted on demand and for reasonable requests.





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




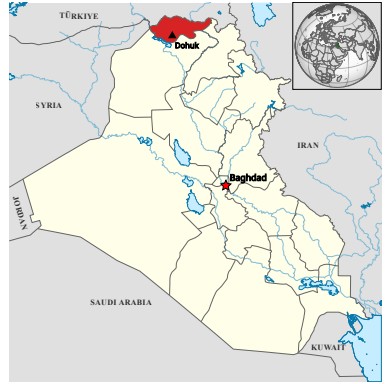

**Figure 1.** Location of Dohuk governorat in the Republic of Iraq (Wikimedia commons). Realised with *QGIS 3.34.5*.





**Figure 2.** Workflow of the soil properties maps production based on Malone et al. 2022 protocol. Realised with *Inkscape 1.4*, and inspired by Shi et al. 2025 design.



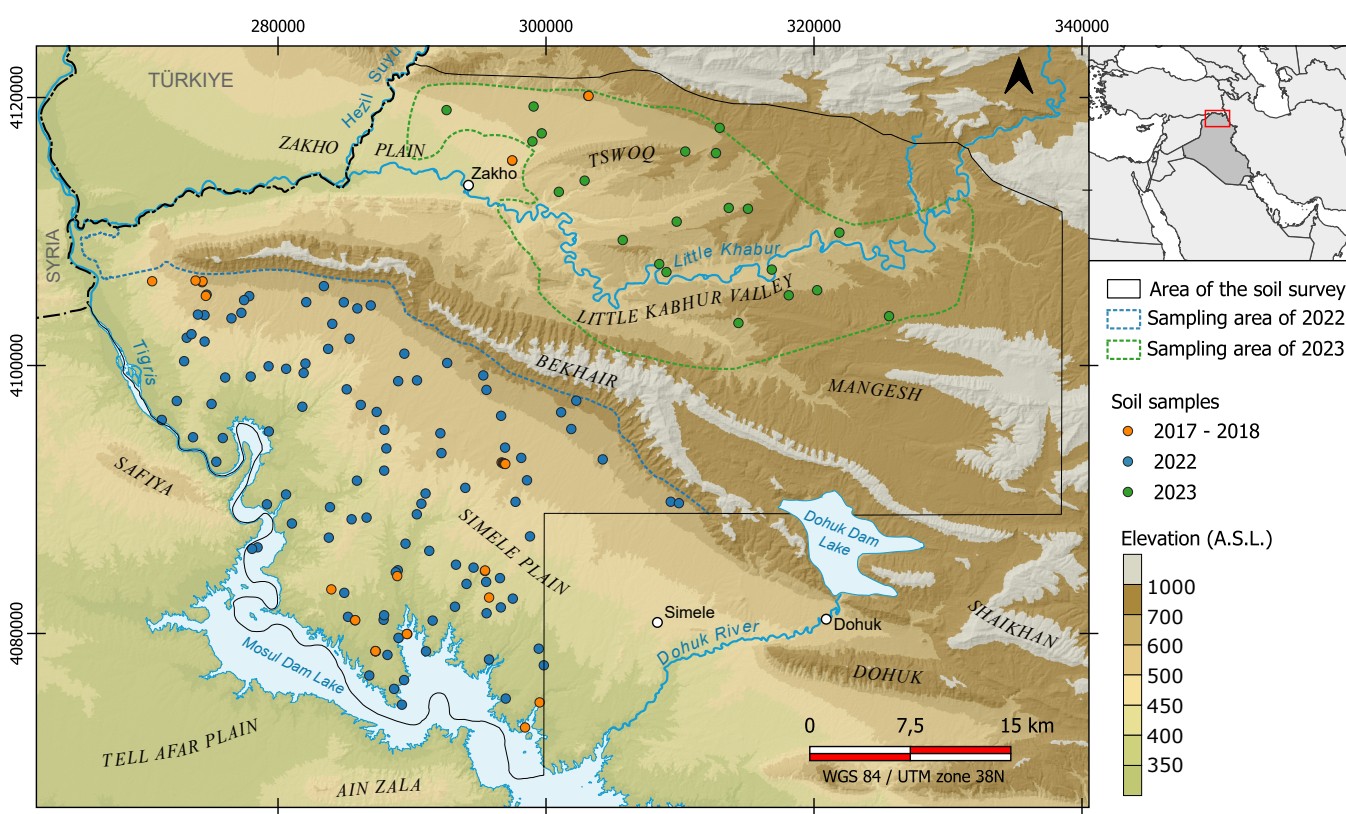

**Figure 3.** Map of the sampling areas and different sample locations (Background: Copernicus data ESA and Airbus 2022). Realised with *QGIS 3.34.5* and *Inkscape 1.4*.



**Figure 4.** Geological map of the studied area compiled from: Ponikarov and Mikhailov 1986; Sissakian et al. 1995; Isiker et al. 2002; Al-Mousawi et al. 2007; Sissakian and Al-Jiburi 2012, 2014 and Doski and McClay 2022 (**LF** = Low-folded zone; **HF** = High-folded zone). Realised with *QGIS 3.34.5* and *Inkscape 1.4*.

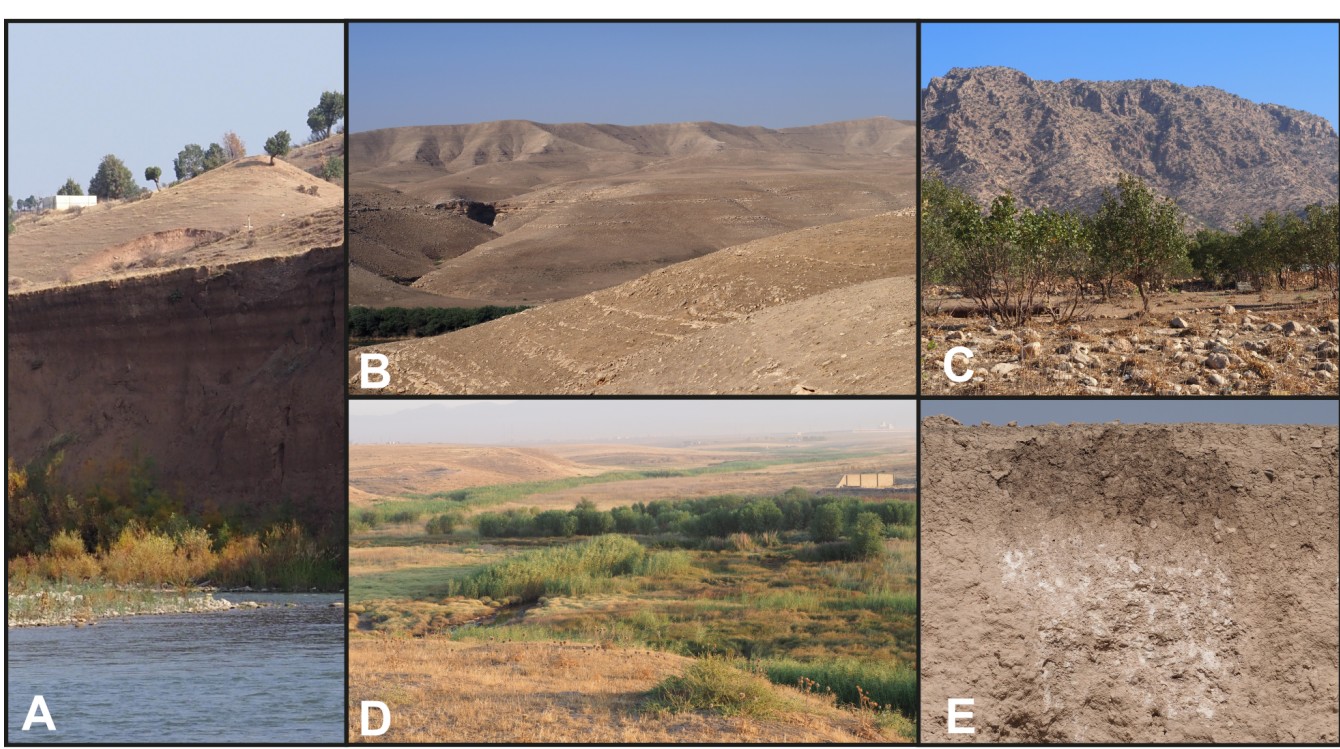

**Figure 5.** Examples of landscapes (A) Terraces of the Little Khabur (10 - 11 m) featuring a succession of colluvial and flood deposits. 37º05'14.46" N 42º56'28.32". (B) Hill and badland landscape on marl formation. *Wādīs* shape this landscape mainly used for grazing. 36º57'16.86" N 42º28'38.53" E. (C) Foothills landscape at the base of the Bekhair. Stones are visible on the surface, and olive trees are cultivated in these foothills. Lithosols, Cambisols or Calcisols dominate this landscape. 37º03'50.73" N 42º34'50.71" E. (D) *Wādī* landscape with heavily developed vegetation, shrubs and small trees. Fluvisols or Vertisols are usually associated with this environment. 36º54'35.33" N 42º44'29.64" E. (E) Calcisol developed on a conglomerate formation formation. The top 10-15 cm shows an A humic horizon, followed by 5-10 cm of a Bt horizon and a C calcitic horizon at the bottom. 37º01'24.61" N 42º30'37.11" E. Realised with *Inkscape 1.4*.





**Figure 6.** Soil type map based on the WRB 2006 classification. Observations come from survey informations and previous work of Buringh 1957 and Altaie 1968. Realised with *QGIS 3.34.5* and *Inkscape 1.4*.



**Table 1.** Sampling campaigns.

| Year | Number of sites | Number of samples | Samples selected for measuring soil properties | Depth explored | Area surveyed |
|---|---|---|---|---|---|
| 2017 - 2018 | 16 | 29 | 29 | 0 - 10 cm | - |
| 2022 | 101 | 445 | 50 | 0 - 100 cm | 830 km$^2$ |
| 2023 | 21 | 87 | 29 | 0 - 100 cm | [1]490 km$^2$ |
| Total for soil properties prediction | 138 | 560 [2] | 108 | 0 - 100 cm | - |
| **Total for DSM** | 122 | 531[3] | - | 0 - 100 cm | 2,280 km$^2$ |

[1] Original size of the non-reduced area is 1450 km$^2$.

[2] One sample did have FTIR spectra out of range, and therefore was not used.

[3] Samples of the 2017 - 2018 campaigns were not used for the DSM due to the absence of several depth increments.



**Table 2.** Descriptive statistics of soil properties observed and predicted.

| | pH | CaCo$_3$ [%] | N$_t$ [%] | C$_t$ [%] | Corg [%] | EC [µS/cm] | MWD [mm] | Sand [%] | Silt [%] | Clay [%] |
|---|---|---|---|---|---|---|---|---|---|---|
| | | | | Observed values | | | | | | |
| Minimum | 6.93 | 3.61 | -[4] | 1.89 | 0 | 90 | 0.01 | 2.24 | 2.3 | 8.5 |
| Maximum | 8.2 | 84.27 | 0.67 | 10.75 | 7.65 | 932 | 0.4091 | 63.2 | 65.5 | 67.4 |
| Mean | 7.29 | 30.61 | 0.12 | 4.79 | 1.11 | 287.20 | 0.1 | 19.38 | 44.02 | 36.48 |
| Q1 | 7.14 | 23.91 | 0.07 | 3.76 | 0.49 | 200.67 | 0.04 | 7.65 | 36.12 | 25.62 |
| Q3 | 7.4 | 34.28 | 0.14 | 5.19 | 1.55 | 311 | 0.13 | 27.94 | 51.1 | 47.2 |
| Std. deviation | 0.2 | 12.48 | 0.09 | 1.72 | 0.98 | 158.62 | 0.07 | 15.34 | 10.55 | 14.45 |
| Skewness | 1.23 | 1.47 | 3.16 | 1.49 | 3.23 | 2.43 | 1.42 | 1.13 | -0.55 | 0.064 |
| | | | | Predicted values | | | | | | |
| Minimum | 6.96 | 3.82 | 0.022 | 1.82 | 0.12 | 136.6 | 0.018 | 1.42 | 25.53 | 10.96 |
| Maximum | 7.75 | 87.53 | 0.64 | 10.69 | 7.02 | 345.36 | 0.241 | 62.4 | 58.92 | 59.06 |
| Mean | 7.23 | 31.46 | 0.078 | 4.5 | 0.83 | 249.9 | 0.073 | 14.7 | 45.95 | 38.39 |
| Q1 | 7.16 | 25.85 | 0.05 | 3.82 | 0.49 | 231.84 | 0.047 | 7.43 | 42.93 | 33.2 |
| Q3 | 7.28 | 35 | 0.091 | 4.92 | 0.98 | 269.94 | 0.089 | 18.09 | 49.35 | 44.84 |
| Std. deviation | 0.11 | 10.09 | 0.049 | 1.28 | 0.57 | 30.24 | 0.036 | 10.81 | 5.84 | 9.36 |
| Skewness | 1.03 | 1.59 | 4.58 | 1.38 | 3.89 | -0.43 | 1.28 | 1.71 | -0.61 | 0.75 |

[4] Device could not measure concentration below 0.03.



**Table 3.** Environmental covariates by soil forming factor, for the digital soil mapping. These factors are based on *Scorpan* model (**Equation 2**; McBratney et al. 2003; **NIR** = Near-infrared; **NDVI** = Normalised difference vegetation index; **SWIR** = Short wavelength infrared; **EVI** = Enhanced vegetation index; **SAVI** = Soil adjusted vegetation index; **NDMI** = Normalised difference moisture index; **CORSI** = Combined spectral response index; **LST** = Land surface temperature; **TVI** = Transformed vegetation index; **LSWI** = Land surface water index; **DEM** = Digital elevation model; **MrRTF** = Multiresolution index of the ridge top flatness; **MrVBF** = Multiresolution index of the valley bottom flatness; **TPI** = Topographic position index; **TWI** = Topographic wetness index) .

| Code | Factor | Covariates |
|---|---|---|
| LA.17 - LA.20 | Soil ($s$) | Landsat 8 clay, salinity, gypsum and carbonate indexes |
| LA.5, LA.8 - LA.9 | | Landsat 8 NIR and SWIR bands |
| LA.14 | | Landsat 8 CORSI |
| SE.16, SE.21 - SE.23 | | Sentinel 2 clay, salinity, gypsum and carbonate indexes |
| SE.5, SE.7 - SE.11 | | Sentinel 2 NIR, RedEdge and SWIR bands |
| SE.18 | | Sentinel 2 CORSI |
| MO.5 | | Modis NIR band |
| OT.4 | | Landuses map |
| MO.2 - MO.3 | Climate ($c$) | Modis land surface temperature by night and day |
| OT.5 | | Potential evapotranspiration |
| OT.6 | | Precipitation |
| OT.7 | | Solar radiation |
| OT.8 | | Difference between max. and min. temperature |
| OT.9 | | Wind speed |
| LA.1 - LA.2, LA.6 - LA.7 | Organisms ($o$) | Landsat 8 blue, green panchromatic and red bands |
| LA.3 - LA.4 | | Landsat 8 NDVI and NDWI |
| LA.10 - LA.13, LA.15 | | Landsat 8 EVI, SAVI, TVI, NDMI and LSWI |
| MO.6 | | Modis red band |
| MO.4 | | Modis NDVI |
| MO.1, MO.7 - MO.8 | | Modis EVI, TVI and SAVI |
| SE.1 - SE.2, SE. 6, SE.12 | | Sentinel 2 blue, green red and water vapor bands |
| SE.3 - SE.4 | | Sentinel 2 NDVI and NDWI |
| SE.13 - SE.15, SE.17, SE.19 | | Sentinel 2 EVI, SAVI, TVI, NDMI and LSWI |
| TE.5 | Relief ($r$) | DEM |
| TE.1 | | Aspect |
| TE.2 - TE.3, TE.6, TE.23 - TE.24 | | Channel network base level and distance, flow accumulation, total catchment area and valley depth |
| TE.4, TE.10, TE.13 | | Convexity, negative and positive openness |
| TE.7, TE.12, TE.14 | | General, plan and profile curvature |
| TE.8 - TE.9 | | MrRTF and MrVBF |
| TE.11, TE.17 | | Normalised and standardised height |
| TE.15 - TE.16 | | Slope height and slope |
| TE.21 - TE.22 | | TPI and TWI |
| LA.16 | Parent material ($p$) | Landsat 8 brightness index |
| SE.18 | | Sentinel 2 brightness index |
| MO.9 | Modis brightness index | |
| OT.3 | | Geology |
| OT.3 | Age ($a$) | Geomorphology |
| TE.18 | | Surface landform |
| TE.19 - TE.20 | | Terrain ruggedness index and texture |
| OT.1 | Space ($n$) | Distance to rivers |





**Table 4.** Cubist model evaluation metrics.

| | pH | CaCo$_3$ | N$_t$ | C$_t$ | Corg | EC | Sand | Silt | Clay | MWD |
|---|---|---|---|---|---|---|---|---|---|---|
| Spectra | Raw | SG 2.11 | SNV-SG | SNV-SG | SNV-SG | SNV-SG | SNV-SG | SNV-SG | SG 2.11 | Raw |
| MSE | 0.02 | 14.99 | 0.001 | 0.25 | 0.37 | 3507.48 | 44.12 | 68.47 | 58.60 | 0.003 |
| MAE | 0.012 | 2.61 | 0.02 | 0.31 | 0.37 | 45 | 4.75 | 6.21 | 5.99 | 0.04 |
| R$^2$ | 0.44 | 0.90 | 0.83 | 0.91 | 0.61 | 0.29 | 0.81 | 0.38 | 0.72 | 0.47 |
| RMSE | 0.14 | 3.88 | 0.04 | 0.5 | 0.6 | 59.22 | 6.64 | 8.27 | 7.64 | 0.05 |
| CCC | 0.6 | 0.95 | 0.89 | 0.96 | 0.76 | 0.49 | 0.89 | 0.56 | 0.83 | 0.63 |
| RPIQ | 1.24 | 2.77 | 1.71 | 2.72 | 1.15 | 0.93 | 2.66 | 0.98 | 2.43 | 1.25 |

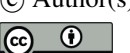

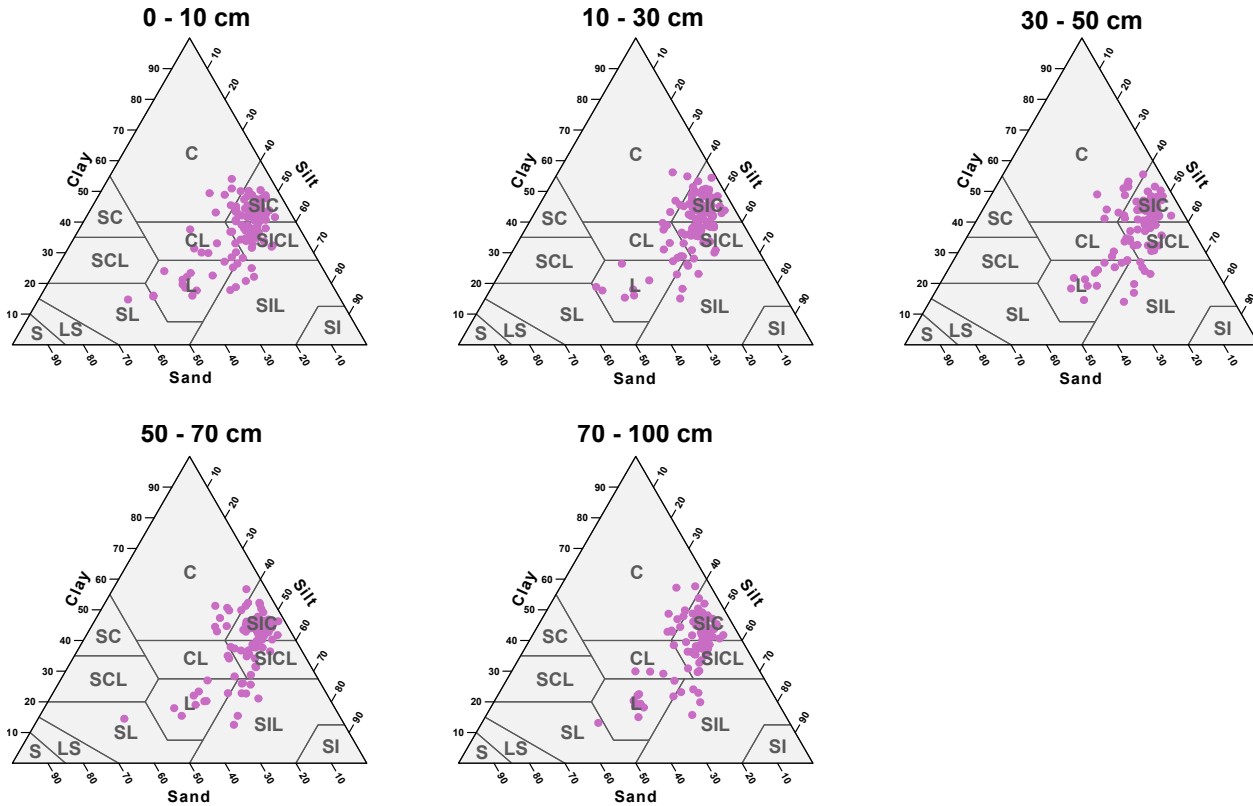

**Figure 7.** Particle size soil predictions, representation in a triangle diagram, according to USDA classification system (WRB, 2006), for each depth increments. **C**: clay; **SC**: sandy clay; **SCL**: sandy clay loam; **CL**: clay loam; **SIC**: silty clay; **SICL**: silty clay loam; **L**: loam; **SIL**: silty loam; **SI**: silt; **SL**: sandy loam; **LS**: loamy sand; **S**: sand. Realised with *R 4.4.0*.





**Table 5.** Soil properties mapping models evaluation metrics.

| Variable | | 0-10 cm | | 10-30cm | | 30-50 cm | | 50-70 cm | | 70-100 cm | |
|---|---|---|---|---|---|---|---|---|---|---|---|
| | | Training | Test | Training | Test | Training | Test | Training | Test | Training | Test |
| pH | Model | QRF | | QRF | | QRF | | QRF | | Knn | |
| | RMSE | 0.09 | 0.09 | 0.09 | 0.09 | 0.09 | 0.1 | 0.09 | 0.16 | 0.09 | 0.11 |
| | $R^2$ | 0.28 | 0.13 | 0.33 | 0.26 | 0.37 | 0.49 | 0.26 | 0.08 | 0.3 | 0.29 |
| | MAE | 0.07 | 0.08 | 0.07 | 0.07 | 0.07 | 0.08 | 0.08 | 0.12 | 0.07 | 0.07 |
| | CCC | - | 0.27 | - | 0.42 | - | 0.5 | - | 0.22 | - | 0.33 |
| | PICP | - | 83.3 | - | 90 | - | 90 | - | 72.2 | - | - |
| CaCo$_3$ | Model | QRF | | Ensemble | | QRF | | SVMr | | Ensemble | |
| | RMSE | 7.23 | 6.72 | 7.59 | 4.99 | 9.01 | 9.39 | 8.55 | 9.84 | 8.15 | 9.97 |
| | $R^2$ | 0.22 | 0.48 | 0.25 | 0.63 | 0.41 | 0.3 | 0.44 | 0.06 | 0.29 | 0.46 |
| | MAE | 5.23 | 5.05 | 5.66 | 3.89 | 6.36 | 5.85 | 6.13 | 7.29 | 6.26 | 7.71 |
| | CCC | - | 0.37 | - | 0.70 | - | 0.27 | - | 0.17 | - | 0.49 |
| | PICP | - | 83.3 | - | - | - | 80 | - | - | - | - |
| N$_t$ | Model | QRF | | QRF | | Knn | | QRF | | Ensemble | |
| | RMSE | 0.05 | 0.04 | 0.02 | 0.03 | 0.02 | 0.02 | 0.3 | 0.026 | 0.02 | 0.02 |
| | $R^2$ | 0.59 | 0.04 | 0.32 | 0.2 | 0.26 | 0.19 | 0.22 | 0.32 | 0.28 | 0.08 |
| | MAE | 0.03 | 0.03 | 0.02 | 0.02 | 0.015 | 0.01 | 0.02 | 0.02 | 0.01 | 0.01 |
| | CCC | - | 0.19 | - | 0.38 | - | 0.38 | - | 0.45 | - | 0.24 |
| | PICP | - | 70.8 | - | 85 | - | - | - | 55.5 | - | - |
| C$_t$ | Model | QRF | | QRF | | Ensemble | | QRF | | CART | |
| | RMSE | 1.14 | 0.82 | 1.01 | 1.08 | 1.19 | 1.03 | 1.09 | 0.92 | 1 | 1.25 |
| | $R^2$ | 0.29 | 0.32 | 0.38 | 0.39 | 0.3 | 0.19 | 0.33 | 0.58 | - | 0.33 |
| | MAE | 0.78 | 0.65 | 0.7 | 0.74 | 0.9 | 0.73 | 0.77 | 0.69 | 0.81 | 0.83 |
| | CCC | - | 0.38 | - | 0.45 | - | 0.42 | - | 0.38 | - | 0.39 |
| | PICP | - | 83.3 | - | 95 | - | - | - | 94.4 | - | - |
| Corg | Model | Cubist | | QRF | | QRF | | Cubist | | Ensemble | |
| | RMSE | 0.72 | 0.38 | 0.34 | 0.45 | 0.25 | 0.29 | 0.23 | 0.28 | 0.22 | 0.23 |
| | $R^2$ | 0.43 | 0.32 | 0.39 | 0.11 | 0.33 | 0.11 | 0.45 | 0.39 | 0.43 | 0.16 |
| | MAE | 0.47 | 0.30 | 0.26 | 0.35 | 0.20 | 0.23 | 0.18 | 0.21 | 0.17 | 0.19 |
| | CCC | - | 0.55 | - | 0.25 | - | 0.29 | - | 0.47 | - | 0.39 |
| | PICP | - | - | - | 75 | - | 70 | - | - | - | - |
| EC | Model | SVMr | | Cubist | | QRF | | Cubist | | CART | |
| | RMSE | 28.91 | 26.3 | 23.35 | 25.05 | 21.92 | 28.68 | 26.59 | 28.41 | 27.6 | 25.67 |
| | $R^2$ | 0.19 | 0.25 | 0.26 | 0.43 | 0.44 | 0.12 | 0.29 | 0.37 | - | 0.26 |
| | MAE | 22 | 21.7 | 18.85 | 16.06 | 17.17 | 22.98 | 20.99 | 23.12 | 22.64 | 19.25 |
| | CCC | - | 0.34 | - | 0.55 | - | 0.22 | - | 0.49 | - | 0.41 |
| | PICP | - | 66.7 | - | - | - | 80 | - | - | - | - |



| Variable | | 0-10 cm | | 10-30cm | | 30-50 cm | | 50-70 cm | | 70-100 cm | |
|---|---|---|---|---|---|---|---|---|---|---|---|
| | | Training | Test | Training | Test | Training | Test | Training | Test | Training | Test |
| MWD | Model | Knn | | Cubist | | QRF | | CART | | QRF | |
| | RMSE | 0.03 | 0.03 | 0.03 | 0.03 | 0.03 | 0.04 | 0.03 | 0.03 | 0.03 | 0.04 |
| | $R^2$ | 0.41 | 0.18 | 0.26 | 0.5 | 0.36 | 0.43 | - | 0.18 | 0.35 | 0.05 |
| | MAE | 0.02 | 0.02 | 0.02 | 0.02 | 0.02 | 0.027 | 0.02 | 0.019 | 0.02 | 0.23 |
| | CCC | - | 0.31 | - | 0.6 | - | 0.38 | - | 0.28 | - | 0.12 |
| | PICP | - | - | - | - | - | 75 | - | - | - | 93.75 |
| alr_sand | Model | QRF | | QRF | | QRF | | CART | | Cubist | |
| | RMSE | 0.48 | 0.68 | 0.69 | 0.83 | 0.74 | 0.86 | 0.88 | 0.94 | 0.77 | 0.65 |
| | $R^2$ | 0.21 | 0.33 | 0.46 | 0.6 | 0.41 | 0.28 | - | 0.36 | 0.39 | 0.26 |
| | MAE | 0.67 | 0.5 | 0.55 | 0.58 | 0.6 | 0.74 | 0.7 | 0.75 | 0.6 | 0.55 |
| | CCC | - | 0.48 | - | 0.45 | - | 0.37 | - | 0.52 | - | 0.43 |
| | PICP | - | 91.66 | - | 90 | - | 85 | - | - | - | - |
| alr_silt | Model | Ensemble | | Ensemble | | QRF | | QRF | | QRF | |
| | RMSE | 0.27 | 0.2 | 0.24 | 0.3 | 0.29 | 0.28 | 0.31 | 0.35 | 0.29 | 0.31 |
| | $R^2$ | 0.32 | 0.49 | 0.32 | 0.46 | 0.37 | 0.19 | 0.35 | 0.14 | 0.31 | 0.13 |
| | MAE | 0.2 | 0.16 | 0.19 | 0.22 | 0.21 | 0.23 | 0.22 | 0.26 | 0.22 | 0.24 |
| | CCC | - | 0.48 | - | 0.43 | - | 0.38 | - | 0.3 | - | 0.23 |
| | PICP | - | - | - | - | - | 90 | - | 66.6 | - | 75 |
| Sand | Model | QRF | | QRF | | QRF | | CART | | Cubist | |
| | RMSE | - | 15.21 | - | 7.33 | - | 7.43 | - | 7.93 | - | 8.87 |
| | $R^2$ | - | 0.05 | - | 0.17 | - | 0.19 | - | 0.25 | - | 0.36 |
| | MAE | - | 9.8 | - | 5.26 | - | 5.34 | - | 5.94 | - | 6.63 |
| | CCC | - | 0.13 | - | 0.29 | - | 0.38 | - | 0.48 | - | 0.42 |
| | PICP | - | 70.8 | - | 95 | - | 85 | - | - | - | 87.5 |
| Silt | Model | Ensemble | | Ensemble | | QRF | | QRF | | QRF | |
| | RMSE | - | 6.49 | - | 7.45 | - | 4.31 | - | 8.09 | - | 7.65 |
| | $R^2$ | - | 0.07 | - | 0.02 | - | 0.42 | - | 0.003 | - | 0.002 |
| | MAE | - | 5.49 | - | 6.01 | - | 3.47 | - | 5.98 | - | 6.04 |
| | CCC | - | 0.24 | - | 0.13 | - | 0.63 | - | 0.05 | - | -0.04 |
| | PICP | - | - | - | - | - | 10 | - | 11.1 | - | 25 |
| Clay | Model | QRF/Ensemble | | QRF/Ensemble | | QRF | | CART/QRF | | Cubist/QRF | |
| | RMSE | - | 8.8 | - | 12.71 | - | 8.17 | - | 9.76 | - | 12.05 |
| | $R^2$ | - | 0.14 | - | 0.12 | - | 0.01 | - | 0.16 | - | 0.0002 |
| | MAE | - | 6.87 | - | 9.71 | - | 6.61 | - | 7.53 | - | 9.09 |
| | CCC | - | 0.29 | - | -0.19 | - | 0.08 | - | 0.28 | - | -0.006 |
| | PICP | - | - | - | - | - | 0 | - | - | - | - |





| Variable | | 0-10 cm | | 10-30cm | | 30-50 cm | | 50-70 cm | | 70-100 cm | |
| --- | --- | --- | --- | --- | --- | --- | --- | --- | --- | --- | --- |
| | | Training | Test | Training | Test | Training | Test | Training | Test | Training | Test |
| | | Training | | | | | | Test | | | |
| Depth | Model | | | | QRF | | | | | | |
| | RMSE | 2.62 (cm$^{0.5}$) | | | | | | 2.42 (cm$^{0.5}$) | | | |
| | R$^2$ | 0.53 | | | | | | 0.57 | | | |
| | MAE | 1.79 | | | | | | 1.54 | | | |
| | CCC | - | | | | | | 0.74 | | | |
| | PICP | - | | | | | | 96.87 | | | |





**Table 6.** Number of covariates selected with `Boruta` and top five factors for every soil property.

| Variable | Total number of covariates selected (For each soil depth increment) | Top covariates selected (Top five covariates present in several depth models) |
|---|---|---|
| pH | 8; 11; 11; 7; 7 | Diff. max. and min. temperature, PET, Sentinel 2 water band, Sentinel 2 EVI and Sentinel 2 NDWI |
| $CaCo_3$ | 11; 11; 13; 12; 9 | Landsat 8 SWIR1, Landsat 8 SWIR2, Sentinel 2 SWIR1 and Sentinel 2 SWIR2 |
| Nt | 18; 17; 8; 6; 8 | PET, Solar radiation, Channel network base level and DEM |
| Ct | 18; 12; 11; 13; 11 | Sentinel 2 SWIR1, Sentinel 2 SWIR2, Sentinel 2 green, Landsat 8 SWIR1 and Landsat 8 SWIR2 |
| Corg | 20; 14; 9; 10; 8 | Sentinel 2 COSRI, Sentinel 2 NDVI, Channel network base level, DEM, and Profil curvature |
| EC | 10; 13; 18; 7; 7 | Landuses map, Sentinel 2 NIR and Sentinel 2 RedEdge3 |
| MWD | 11; 14; 19; 16; 10 | Sentinel 2 EVI, Sentinel 2 SAVI, Sentinel 2 NDVI, Sentinel 2 COSRI and PET |
| alr_sand | 20; 20; 17; 13; 14 | Sentinel 2 EVI, Sentinel 2 TVI, Sentinel 2 SAVI, Sentinel 2 COSRI and Sentinel 2 NDVI |
| alr_silt | 20; 19; 11; 8; 11 | Sentinel 2 SWIR1, Sentinel 2 EVI, Sentinel 2 SAVI, Sentinel 2 NDVI |
| Soil depth | 26 [5] | MrVBF, Slope, DEM, LST Jun.-Jul. and MrRTF |

[5]No selection of the covariates was performed (See. supplementary material)





**Table 7.** Comparative statistics of prediction maps with *SoilGrids 2.0* model.

| Variable | Statistic | Top soil | | Sub-soil | | Lower soil | |
|---|---|---|---|---|---|---|---|
| | | Prediction | *SoilGrids 2.0* | Prediction | *SoilGrids 2.0* | Prediction | *SoilGrids 2.0* |
| | | 0 - 30 cm | 0 - 30 cm | 30 - 70 cm | 30 - 60 cm | 70 - 100 cm | 60 - 100 cm |
| pH | Minimum | 8.141 | 5.750 | 8.135 | 5.755 | 8.118 | 5.798 |
| | Maximum | 8.460 | 9.180 | 8.514 | 9.223 | 8.409 | 9.115 |
| | Mean | 8.274 | 8.522 | 8.304 | 8.647 | 8.268 | 8.704 |
| | Q1 | 8.205 | 8.658 | 8.226 | 8.727 | 8.216 | 8.731 |
| | Q3 | 8.337 | 8.859 | 8.374 | 8.859 | 8.311 | 8.863 |
| | Std. deviation | 0.073 | 0.844 | 0.085 | 0.615 | 0.058 | 0.615 |
| $N_t$ | Minimum | 0.062 | 0.019 | 0.041 | 0.013 | 0.036 | 0.012 |
| [%] | Maximum | 0.233 | 0.025 | 0.101 | 0.021 | 0.091 | 0.021 |
| | Mean | 0.115 | 0.023 | 0.069 | 0.017 | 0.062 | 0.016 |
| | Q1 | 0.086 | 0.022 | 0.058 | 0.015 | 0.056 | 0.013 |
| | Q3 | 0.133 | 0.024 | 0.082 | 0.019 | 0.075 | 0.019 |
| | Std. deviation | 0.039 | 0.002 | 0.013 | 0.002 | 0.012 | 0.003 |
| Corg | Minimum | 0.385 | 0.105 | 0.406 | 0.051 | 0.299 | 0.038 |
| [%] | Maximum | 2.954 | 0.162 | 1.115 | 0.094 | 1.02 | 0.078 |
| | Mean | 1.228 | 0.137 | 0.739 | 0.074 | 0.752 | 0.06 |
| | Q1 | 0.928 | 0.126 | 0.617 | 0.062 | 0.5633 | 0.047 |
| | Q3 | 1.399 | 0.149 | 0.842 | 0.086 | 0.969 | 0.074 |
| | Std. deviation | 0.453 | 0.016 | 0.139 | 0.014 | 0.217 | 0.014 |
| Sand | Minimum | 9.87 | 13.08 | 9.12 | 12.74 | 4.05 | 13.06 |
| [%] | Maximum | 29.38 | 18.07 | 29.88 | 17.44 | 59.95 | 17.73 |
| | Mean | 18.35 | 15.06 | 19.95 | 14.72 | 21.12 | 15.07 |
| | Q1 | 14.36 | 14.01 | 16.25 | 13.78 | 15.86 | 14.13 |
| | Q3 | 22.55 | 15.83 | 23.25 | 15.48 | 25.07 | 15.82 |
| | Std. deviation | 4.47 | 1.39 | 5.36 | 1.27 | 4.26 | 1.26 |
| Silt | Minimum | 38.77 | 34.92 | 33.92 | 34.22 | 2.85 | 34.11 |
| [%] | Maximum | 50.2 | 44.79 | 55.44 | 43.99 | 56.75 | 44.03 |
| | Mean | 44.06 | 40.57 | 43.26 | 39.8 | 43.35 | 39.76 |
| | Q1 | 42.64 | 39.5 | 41.93 | 38.74 | 41.44 | 38.58 |
| | Q3 | 45.53 | 41.07 | 44.75 | 41.05 | 45.36 | 41.11 |
| | Std. deviation | 1.77 | 1.788 | 5.76 | 1.88 | 2.25 | 2 |
| Clay | Minimum | 30.98 | 37.17 | 26.4 | 38.33 | 14.37 | 38.08 |
| [%] | Maximum | 50.2 | 48.1 | 49.47 | 49.41 | 47.54 | 49.11 |
| | Mean | 37.59 | 43.93 | 36.79 | 44.98 | 35.53 | 44.65 |
| | Q1 | 34.50 | 42.88 | 33.48 | 43.92 | 30.99 | 43.59 |
| | Q3 | 40.3 | 45.34 | 48 | 40.03 | 40.16 | 46.01 |
| | Std. deviation | 3.63 | 2.16 | 2.29 | 3.96 | 5.32 | 2.1 |



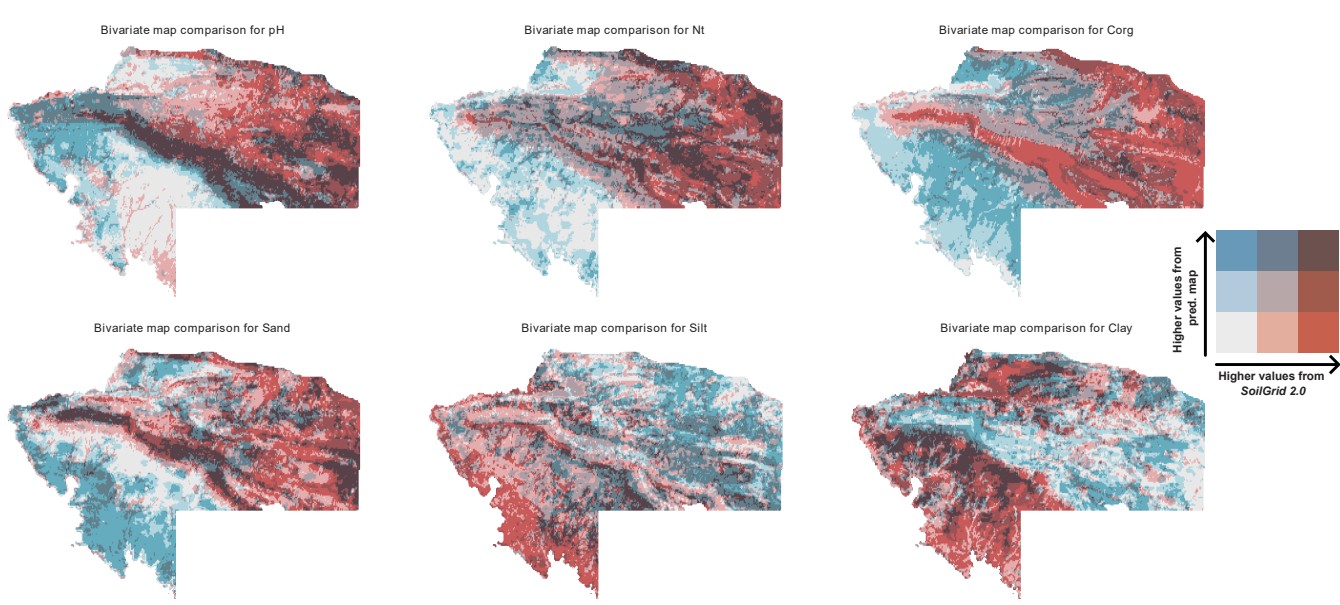

**Figure 8.** Bivariate map of pH, Corg, $N_t$ and texture from prediction maps vs. *SoilGrid 2.0*. Realised with *R 4.4.0*.



**Figure 9.** Prediction maps for the 0 - 10 cm depth increment. Realised with *R 4.4.0*.



**Figure 10.** Prediction maps for the 10 - 30 cm depth increment. Realised with *R 4.4.0*.



**Figure 11.** Prediction maps for the 30 - 50 cm depth increment. Realised with *R 4.4.0.*



**Figure 12.** Prediction maps for the 50 - 70 cm depth increment. Realised with *R 4.4.0*.

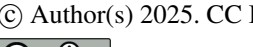

**Figure 13.** Prediction maps for the 70 - 100 cm depth increment. Realised with *R 4.4.0*.

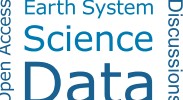

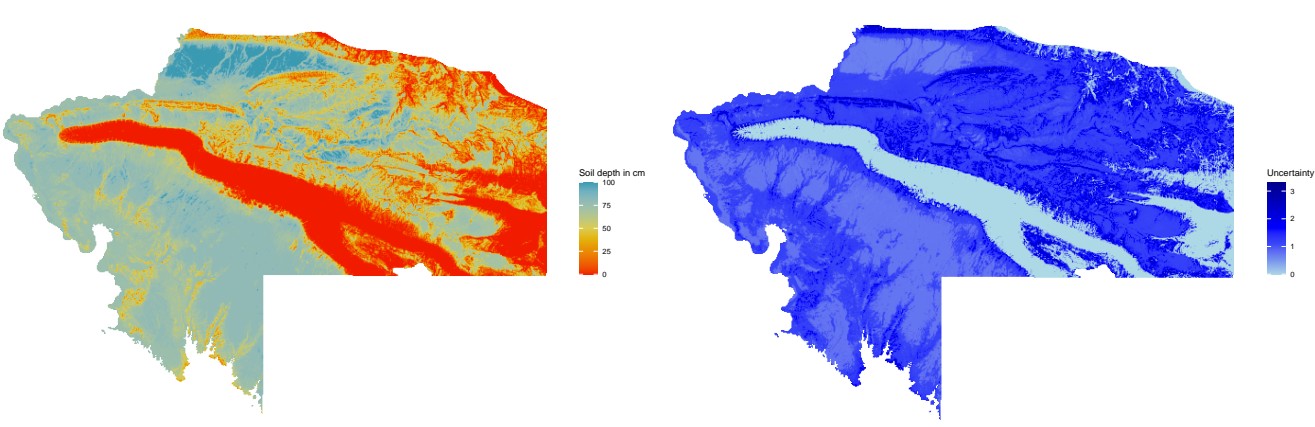

**Figure 14. Left:** Soil depth prediction map. **Right:** Soil depth prediction uncertainty map. Realised with *R 4.4.0*.



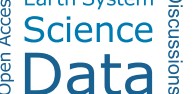

## 950 Appendix A: Geomorphological map

**Figure A1.** Geomorphological map from Forti et al. 2021. Realised with *QGIS 3.34.5* and *Inkscape 1.4*.



## Appendix B: Covariates description and features selection

| Name | Code | Type | Units | Used for | Size (m) | Source |
|---|---|---|---|---|---|---|
| Landsat 8 Blue | LA.1 | Cont. | 0.45 - 0.51 µm | DSM | 30 | EROS 2020 |
| Landsat 8 Green | LA.2 | Cont. | 0.53 - 0.59 µm | CLHs; DSM | 30 | EROS 2020 |
| Landsat 8 NDVI | LA.3 | Cont. | $\frac{NIR-Red}{NIR+Red}$ | DSM | 30 | Rouse et al. 1974 |
| Landsat 8 NDWI | LA.4 | Cont. | $\frac{Green-NIR}{Green+NIR}$ | DSM | 30 | McFeeters 1996 |
| Landsat 8 NIR | LA.5 | Cont. | 0.85 - 0.88 µm | CLHs; DSM | 30 | EROS 2020 |
| Landsat 8 panchromatic | LA.6 | Cont. | 0.52 - 0.90 µm | DSM | 15 | EROS 2020 |
| Landsat 8 Red | LA.7 | Cont. | 0.64 - 0.67 µm | CLHs; DSM | 30 | EROS 2020 |
| Landsat 8 SWIR1 | LA.8 | Cont. | 1.57 - 1.65 µm | CLHs; DSM | 30 | EROS 2020 |
| Landsat 8 SWIR2 | LA.9 | Cont. | 2.11 - 2.29 µm | CLHs; DSM | 30 | EROS 2020 |
| Landsat 8 EVI | LA.10 | Cont. | $2.5\frac{NIR-Red}{(NIR+6Red-7.5Blue)+1}$ | DSM | 30 | Huete et al. 1994 |
| Landsat 8 SAVI | LA.11 | Cont. | $1.5\frac{NIR-Red}{NIR+Red+0.5}$ | DSM | 30 | Huete 1988 |
| Landsat 8 TVI | LA.12 | Cont. | $\sqrt{NDVI+0.5}$ | DSM | 30 | Deering 1975 |
| Landsat 8 NDMI | LA.13 | Cont. | $\frac{NIR-SWIR1}{NIR+SWIR1}$ | DSM | 30 | Gao 1996 |
| Landsat 8 CORSI | LA.14 | Cont. | $\frac{Blue+Green}{Red+NIR}NDVI$ | DSM | 30 | Fernández-Buces et al. 2006 |
| Landsat 8 LSWI | LA.15 | Cont. | $\frac{NIR-SWIR1}{NIR+SWIR1}$ | DSM | 30 | Chandrasekar et al. 2010 |
| Landsat 8 Brigthness index | LA.16 | Cont. | $\sqrt{Red^2+NIR^2}$ | DSM | 30 | Khan et al. 2001 |
| Landsat 8 Clay index | LA.17 | Cont. | $\frac{SWIR1}{SWIR2}$ | DSM | 30 | Bousbih et al. 2019 |
| Landsat 8 Salinity index | LA.18 | Cont. | $\frac{SWIR1-SWIR2}{SWIR1-NIR}$ | DSM | 30 | Abuelgasim and Ammad 2019 |
| Landsat 8 Carbonate index | LA.19 | Cont. | $\frac{Red}{Green}$ | DSM | 30 | Boettinger et al. 2008 |
| Landsat 8 Gysum index | LA.20 | Cont. | $\frac{SWIR1-SWIR2}{SWIR1+SWIR2}$ | DSM | 30 | Nield et al. 2007 |
| Landsat 5 Green | LA5.1 | Cont. | 0.52 - 0.60 µm | Soil Depth | 30 | EROS 2013 |
| Landsat 5 Blue | LA5.2 | Cont. | 0.63 - 0.69 µm | Soil Depth | 30 | EROS 2013 |
| Landsat 5 Red | LA5.3 | Cont. | 0.76 - 0.90 µm | Soil Depth | 30 | EROS 2013 |
| Landsat 5 NIR | LA5.4 | Cont. | 2.08 - 2.35 µm | Soil Depth | 30 | EROS 2013 |
| Landsat 5 NDVI | LA5.5 | Cont. | $\frac{NIR-Red}{NIR+Red}$ | Soil Depth | 30 | Rouse et al. 1974 |
| Landsat 5 NDWI | LA5.6 | Cont. | $\frac{Green-NIR}{Green+NIR}$ | Soil Depth | 30 | McFeeters 1996 |
| LST Apr.-May | LST.1 | Cont. | Kelvin | Soil Depth | 1000 | Hulley and Hook 2018 |
| LST Feb.-Mars | LST.2 | Cont. | Kelvin | Soil Depth | 1000 | Hulley and Hook 2018 |
| LST Jun.-Jul. | LST.3 | Cont. | Kelvin | Soil Depth | 1000 | Hulley and Hook 2018 |
| LST Oct.-Nov. | LST.4 | Cont. | Kelvin | Soil Depth | 1000 | Hulley and Hook 2018 |
| MODIS EVI | MO.1 | Cont. | $2.5\frac{NIR-Red}{(NIR+6Red-7.5Blue)+1}$ | DSM | 250 | Huete et al. 1994 |
| MODIS LST day | MO.2 | Cont. | Kelvin[6] | DSM | 1000 | Wan et al. 2021 |
| MODIS LST night | MO.3 | Cont. | Kelvin[6] | DSM | 1000 | Wan et al. 2021 |
| MODIS NDVI | MO.4 | Cont. | $\frac{NIR-Red}{NIR+Red}$ | DSM | 250 | Didan 2021 |
| MODIS NIR | MO.5 | Cont. | 0.841 - 0.876 µm | DSM | 250 | Vermote 2021 |

[6]Converted into °C





| Name | Code | Type | Unit/Formula | Use | Res. | Source |
|---|---|---|---|---|---|---|
| MODIS Red | MO.6 | Cont. | 0.62 - 0.67 µm | DSM | 250 | Vermote 2021 |
| MODIS SAVI | MO.7 | Cont. | $1.5\frac{NIR-Red}{NIR+Red+0.5}$ | DSM | 250 | Huete 1988 |
| MODIS TVI | MO.8 | Cont. | $\sqrt{NDVI+0.5}$ | DSM | 250 | Deering 1975 |
| MODIS Brightness index | MO.9 | Cont. | $\sqrt{Red^2+NIR^2}$ | DSM | 250 | Khan et al. 2001 |
| Distance rivers | OT.1 | Cont. | Meters | DSM | 25 | ESA and Airbus 2022 |
| Geology | OT.2 | Dis. | 35 class | CLHs; DSM; Soil Depth | -[7] | Sissakian et al. 1995; Al-Mousawi et al. 2007 |
| Geomorphology | OT.3 | Dis. | 17 class | CLHs; DSM; Soil Depth | -??? | Forti et al. 2021 |
| Landuses | OT.4 | Dis. | 11 class | RUSLE; DSM; Soil Depth | 10 | Zanaga et al. 2021 |
| Potential evapotranspiration | OT.5 | Cont. | mm | DSM; Soil Depth | 750 | Zomer and Trabucco, 2022; Zomer et al., 2022 |
| Precipitation | OT.6 | Cont. | mm | RUSLE; DSM; Soil Depth | 1000 | Fick and Hijmans 2017 |
| Solar radiation | OT.7 | Cont. | Kj m-2 | DSM; Soil Depth | 1000 | Fick and Hijmans 2017 |
| Diff. max. and min. temperature | OT.8 | Cont. | °C | DSM | 1000 | Fick and Hijmans 2017 |
| Wind speed | OT.9 | Cont. | m s-1 | DSM; Soil Depth | 1000 | Fick and Hijmans 2017 |
| Temperature average | OT.10 | Cont. | °C | Soil Depth | 1000 | Fick and Hijmans 2017 |
| RUSLE | OT.11 | Cont. | $Mg/ha^{-1}$ per $year^{-1}$ | CLHs | 25 | Mathias Bellat |
| Soil estimation | OT.12 | Cont. | - | CLHs | 30 | Nafiseh Kakhani |
| HWSD V2 | OT.13 | Cont. | - | RUSLE | 1000 | FAO and IIASA 2023 |
| Sentinel 2 Blue | SE.1 | Cont. | 0.492 - 0.496 µm | DSM | 10 | ESA 2022 |
| Sentinel 2 Green | SE.2 | Cont. | 0.559 - 0.560 µm | DSM | 10 | ESA 2022 |
| Sentinel 2 NDVI | SE.3 | Cont. | $\frac{NIR-Red}{NIR+Red}$ | DSM | 20 | Rouse et al. 1974 |
| Sentinel 2 NDWI | SE.4 | Cont. | $\frac{Green-NIR}{Green+NIR}$ | DSM | 20 | McFeeters 1996 |
| Sentinel 2 NIR | SE.5 | Cont. | 0.833 - 0.835 µm | DSM | 10 | ESA 2022 |
| Sentinel 2 Red | SE.6 | Cont. | 0.664 - 0.665 µm | DSM | 10 | ESA 2022 |
| Sentinel 2 RedEdge1 | SE.7 | Cont. | 0.738 - 0.739 µm | DSM | 20 | ESA 2022 |
| Sentinel 2 RedEdge2 | SE.8 | Cont. | 0.739 - 0.740 µm | DSM | 20 | ESA 2022 |
| Sentinel 2 RedEdge3 | SE.9 | Cont. | 0.779 - 0.782 µm | DSM | 20 | ESA 2022 |
| Sentinel 2 SWIR1 | SE.10 | Cont. | 1.610 - 1.613 µm | DSM | 20 | ESA 2022 |
| Sentinel 2 SWIR2 | SE.11 | Cont. | 2.185 - 2.202 µm | DSM | 20 | ESA 2022 |
| Sentinel 2 Water vapor | SE.12 | Cont. | 0.943 - 0.945 µm | DSM | 90 | ESA 2022 |
| Sentinel 2 EVI | SE.13 | Cont. | $2.5\frac{NIR-Red}{(NIR+6Red-7.5Blue)+1}$ | DSM | 20 | Huete et al. 1994 |
| Sentinel 2 SAVI | SE.14 | Cont. | $1.5\frac{NIR-Red}{NIR+Red+0.5}$ | DSM | 20 | Huete 1988 |
| Sentinel 2 TVI | SE.15 | Cont. | $\sqrt{NDVI+0.5}$ | DSM | 20 | Deering 1975 |
| Sentinel 2 Clay index | SE.16 | Cont. | $\frac{SWIR1}{SWIR2}$ | DSM | 20 | Bousbih et al. 2019 |
| Sentinel 2 NDMI | SE.17 | Cont. | $\frac{NIR-SWIR1}{NIR+SWIR1}$ | DSM | 20 | Gao 1996 |
| Sentinel 2 COSRI | SE.18 | Cont. | $\frac{Blue+Green}{Red+NIR}NDVI$ | DSM | 20 | Fernández-Buces et al. 2006 |
| Sentinel 2 LSWI | SE.19 | Cont. | $\frac{NIR-SWIR1}{NIR+SWIR1}$ | DSM | 20 | Chandrasekar et al. 2010 |

[7]Original map resolution 1 : 250 000.





| | | | | | | |
|---|---|---|---|---|---|---|
| Sentinel 2 Brightness index | SE.20 | Cont. | $\sqrt{Red^2 + NIR^2}$ | DSM | 20 | Khan et al. 2001 |
| Sentinel 2 Salinity index | SE.21 | Cont. | $\frac{SWIR1 - SWIR2}{SWIR1 - NIR}$ | DSM | 20 | Abuelgasim and Ammad 2019 |
| Sentinel 2 Carbonate index | SE.22 | Cont. | $\frac{Red}{Green}$ | DSM | 20 | Boettinger et al. 2008 |
| Sentinel 2 Gysum index | SE.23 | Cont. | $\frac{SWIR1 - SWIR2}{SWIR1 + SWIR2}$ | DSM | 20 | Nield et al. 2007 |
| Aspect | TE.1 | Cont. | Radian | DSM; Soil Depth | 25 | ESA and Airbus 2022 |
| Channel network base level | TE.2 | Cont. | - | DSM | 25 | ESA and Airbus 2022 |
| Channel network distance | TE.3 | Cont. | - | DSM | 25 | ESA and Airbus 2022 |
| Convexity | TE.4 | Cont. | - | DSM | 25 | ESA and Airbus 2022 |
| DEM fill | TE.5 | Cont. | Meters | RUSLE; CLHs; DSM; Soil Depth | 25 | ESA and Airbus 2022 |
| Flow accumulation | TE.6 | Cont. | - | DSM | 25 | ESA and Airbus 2022 |
| General curvature | TE.7 | Cont. | - | DSM; Soil Depth | 25 | ESA and Airbus 2022 |
| MrRTF | TE.8 | Cont. | - | DSM; Soil Depth | 25 | ESA and Airbus 2022 |
| MrVBF | TE.9 | Cont. | - | DSM; Soil Depth | 25 | ESA and Airbus 2022 |
| Negative openness | TE.10 | Cont. | Radian | DSM | 25 | ESA and Airbus 2022 |
| Normalized height | TE.11 | Cont. | - | DSM | 25 | ESA and Airbus 2022 |
| Plan curvature | TE.12 | Cont. | - | DSM; Soil Depth | 25 | ESA and Airbus 2022 |
| Positive openness | TE.13 | Cont. | Radian | DSM | 25 | ESA and Airbus 2022 |
| Profile curvature | TE.14 | Cont. | - | DSM; Soil Depth | 25 | ESA and Airbus 2022 |
| Slope height | TE.15 | Cont. | - | DSM | 25 | ESA and Airbus 2022 |
| Slope | TE.16 | Cont. | Radian | DSM | 25 | ESA and Airbus 2022 |
| Standardized height | TE.17 | Cont. | - | DSM | 25 | ESA and Airbus 2022 |
| Surface landform | TE.18 | Cont. | - | DSM | 25 | ESA and Airbus 2022 |
| Terrain ruggedness Index | TE.19 | Cont. | - | DSM | 25 | ESA and Airbus 2022 |
| Terrain texture | TE.20 | Cont. | - | DSM | 25 | ESA and Airbus 2022 |
| TPI | TE.21 | Cont. | - | DSM; Soil Depth | 25 | ESA and Airbus 2022 |
| TWI | TE.22 | Cont. | - | CLHs | 25 | ESA and Airbus 2022 |
| Total catchment area | TE.23 | Cont. | - | DSM | 25 | ESA and Airbus 2022 |
| Valley depth | TE.24 | Cont. | - | DSM | 25 | ESA and Airbus 2022 |

Table A1: Covariates used for the different process and modelling (**Cont.** = Continuous data; **Dis.** = Discrete data; **CLHs** = Cluster Latin Hypercube sampling; **DSM** = Digital soil mapping; **NDWI** = Normalised difference water index; **NIR** = Near-infrared; **NDVI** = Normalised difference vegetation index; **SWIR** = Short wavelength infrared; **EVI** = Enhanced vegetation index; **SAVI** = Soil adjusted vegetation index; **NDMI** = Normalised difference moisture index; **CORSI** = Combined spectral response index; **LST** = Land surface temperature; **TVI** = Transformed vegetation index; **LSWI** = Land surface water index; **DEM** = Digital elevation model; **MrRTF** = Multiresolution index of the ridge top flatness; **MrVBF** = Multiresolution index of the valley bottom flatness; **TPI** = Topographic position index; **TWI** = Topographic wetness index) .



## Appendix B: Boruta selections

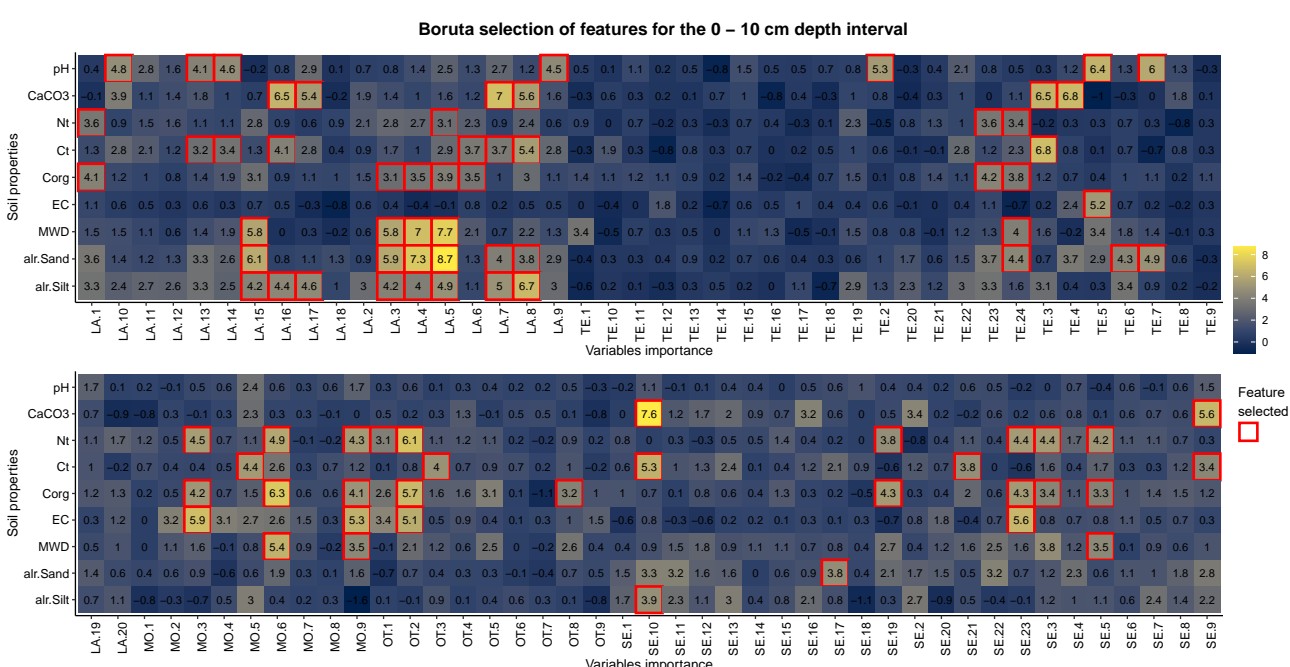

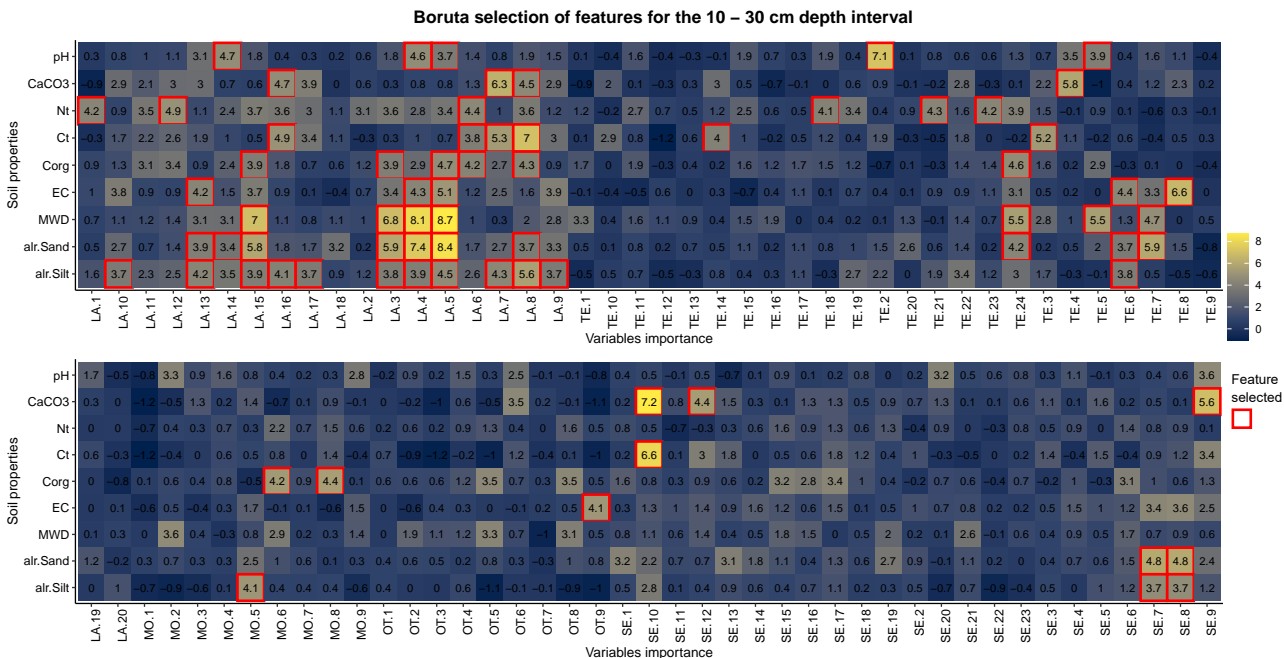

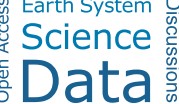

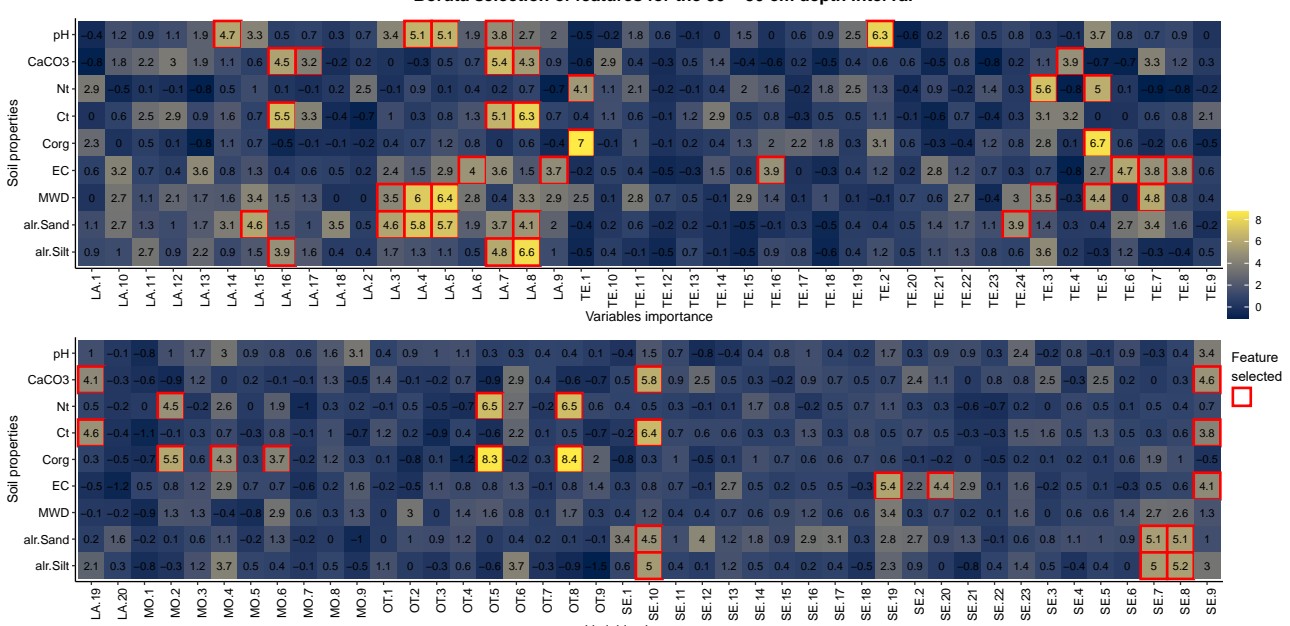

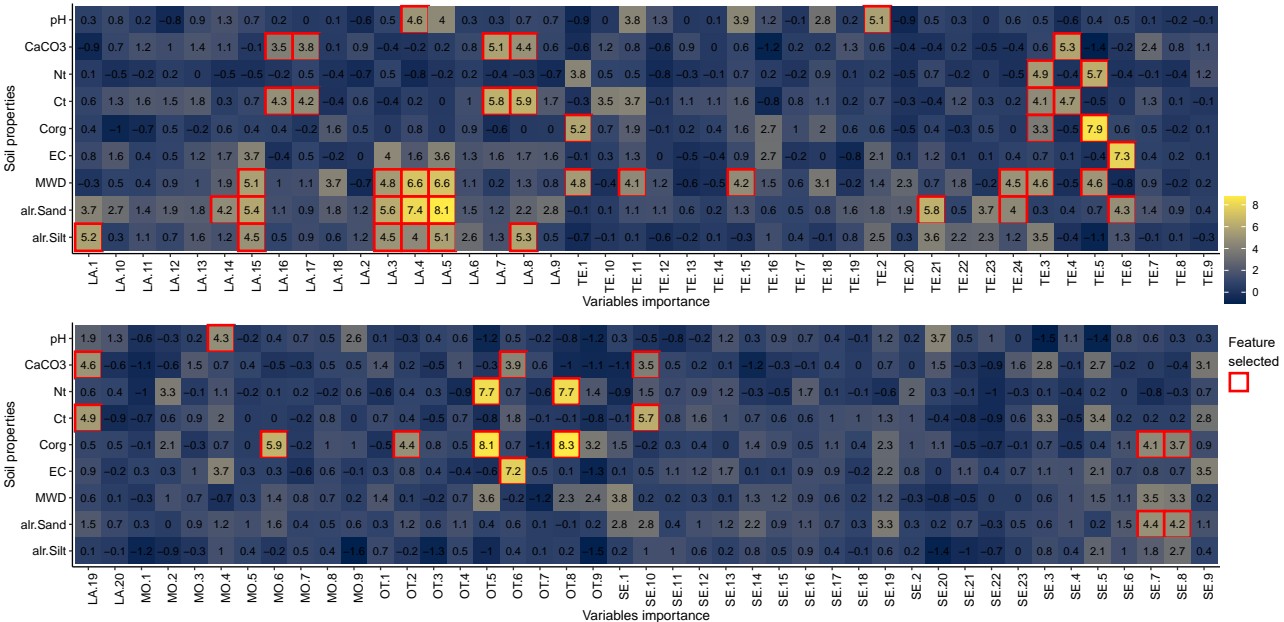



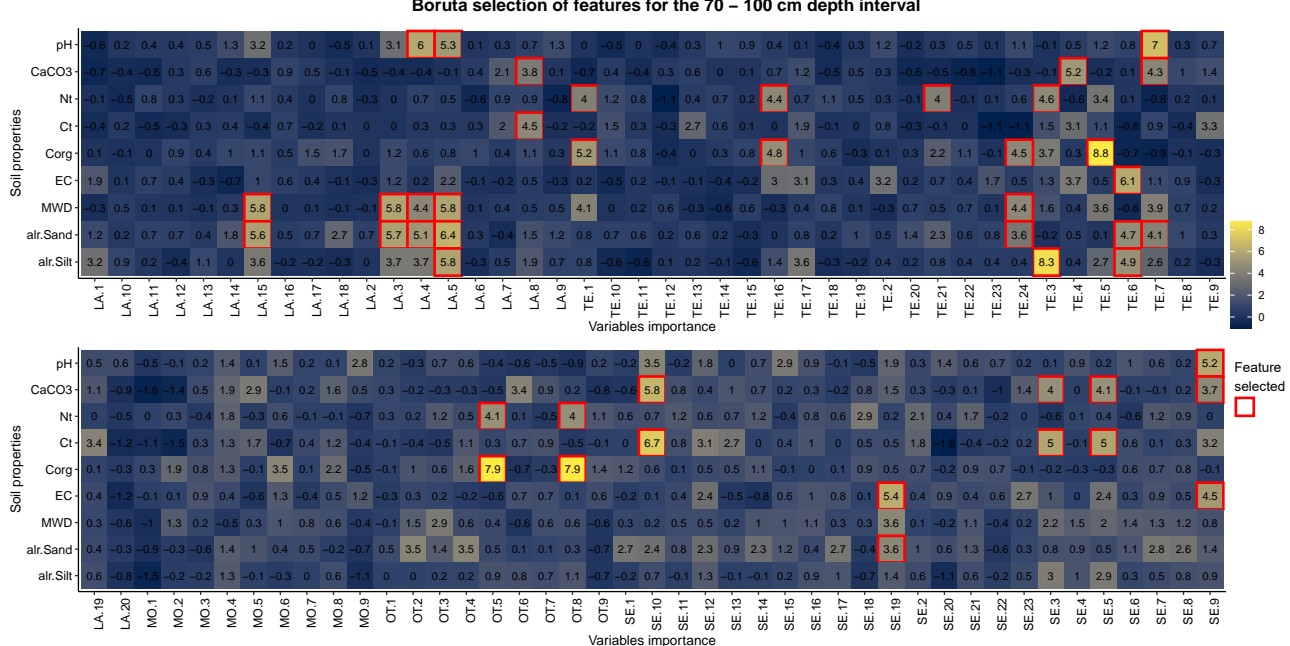

**Figure C1.** Boruta feature selection for each soil property. Realised with *R 4.4.0*.





| Model | Variable | Increment | Parameters | | |
|---|---|---|---|---|---|
| | | | *cp* | | |
| CART | $C_t$ | 70-100cm | 0.1713 | | |
| | EC | 70-100cm | 0.0612 | | |
| | MWD | 50-70cm | 0.0997 | | |
| | alr_sand | 50-70cm | 0.0152 | | |
| | | | *K* | | |
| Knn | pH | 70-100 cm | 7 | | |
| | $N_t$ | 30-50cm | 7 | | |
| | MWD | 0-10cm | 11 | | |
| | | | *sigma* | *C* | |
| SVMr | CaCO$_3$ | 50-70cm | 0.06 | 4 | |
| | EC | 0-10cm | 0.06 | 2 | |
| | | | *committees* | *neighbors* | |
| Cubist | Corg | 0-10cm | 5 | 9 | |
| | Corg | 50-70cm | 1 | 9 | |
| | EC | 10-30cm | 10 | 7 | |
| | EC | 50-70cm | 5 | 9 | |
| | MWD | 10-30cm | 5 | 0 | |
| | alr_sand | 70-100cm | 15 | 9 | |
| | | | *mtry* | | |
| Ensemble | CaCO$_3$ | 10-30cm | 2 | | |
| | CaCO$_3$ | 70-100cm | 4 | | |
| | $N_t$ | 70-100cm | 8 | | |
| | $C_t$ | 30-50cm | 1 | | |
| | Corg | 70-100cm | 1 | | |
| | alr_silt | 0-10cm | 1 | | |
| | alr_silt | 10-30cm | 1 | | |

| Model | Variable | Increment | Parameters | | |
|---|---|---|---|---|---|
| | | | *ntree* | *mtry* | *nodesize* |
| QRF | pH | 0-10 cm | 500 | 2 | - |
| | | 10-30 cm | 500 | 1 | - |
| | | 30-50 cm | 500 | 5 | - |
| | | 50-70 cm | 500 | 5 | - |
| | CaCO$_3$ | 0-10cm | 500 | 1 | - |
| | | 30-50cm | 500 | 1 | - |
| | $N_t$ | 0-10cm | 500 | 10 | - |
| | | 10-30cm | 500 | 3 | - |
| | | 50-70cm | 500 | 1 | - |
| | $C_t$ | 0-10cm | 500 | 9 | - |
| | | 10-30cm | 500 | 1 | - |
| | | 50-70cm | 500 | 2 | - |
| | Corg | 10-30cm | 500 | 2 | - |
| | | 30-50cm | 500 | 1 | - |
| | EC | 30-50cm | 500 | 1 | - |
| | MWD | 30-50cm | 500 | 1 | - |
| | | 70-100cm | 500 | 6 | - |
| | alr_sand | 0-10cm | 500 | 2 | - |
| | | 10-30cm | 500 | 1 | - |
| | | 30-50cm | 500 | 11 | - |
| | alr_silt | 30-50cm | 500 | 2 | - |
| | | 50-70cm | 500 | 2 | - |
| | | 70-100cm | 500 | 2 | - |
| | Depth | - | 500 | 1 | 6 |

**Table E1.** Tuning parameters used for the digital soil mapping of soil properties and depth (*cp* = Complexity parameter; *k* = N° of Neighbors; *C* = Cost; *mtry* = Number of predictors; *ntree* = Number of trees; *nodesize* = Minimum node size).

## Appendix D: Tuning parameters for prediction maps