# Peer review of "Soil information and soil property maps for the Kurdistan region, Dohuk governorate (Iraq)"

_Earth System Science Data, 2025_

## Author Comment (AC1)

Recommender 2 review

Answer to the recommender review

**Changes in the text**

essd-2025-418 "Soil information and soil property maps for the Kurdistan region, Dohuk governorate (Iraq)"

Bellat et al.

**Review by Anonymous Referee #2, 14 Nov 2025**

This manuscript focuses on regional digital soil mapping in Iraq, using 532 soil samples and 85 covariates to produce soil maps via machine learning. While the modeling approaches are generally appropriate, the work falls short in two critical aspects: (1) Limited Geographical Scope: The investigated region is quite small. Consequently, the resulting dataset has limited implications and applicability for the broader scientific community, despite its location is in Iraq. (2) Limited Novelty: The modeling framework adopted is standard practice in digital soil mapping and lacks significant methodological novelty. Given these limitations, specifically the dataset's limited scope and the conventional nature of the modeling, this work does not meet the high standards for originality and impact required for publication in Earth System Science Data.

We sincerely appreciate the time referee 2 took to read the preprint and highlight the adapted modelling approach in our paper. The reviewer identified two critical aspects of our preprint.

- 1) Indeed, the studied area (2,280 km2) is "relatively small" regarding other datasets available in ESSD. However, in other case, regional to local data are also available (e.g. Lorenz et al., 2021; Ardizzone et al., 2023; Błaszczyk et al. 2024). We do think that high-quality regional datasets are necessary to feed and improve other larger datasets. Furthermore, as referee 2 expressed, data on the Iraq region are critically lacking. No regional data set from any kind of observations is available on Iraq in the whole *ESSD* (accessed on 14/11/2025). We do think that underrepresented regions of the globe do need and deserve high-quality, standardised data, as the one proposed in this paper and, more generally, in *ESSD*. Qualitative data presented in the preprint (soil classes map) is also hardly expendable at a large scale, as regional patterns can not always be transposed. Finally, the comparison with the *SoilGrid.2.0* product used in the study also highlights the poor quality of such global products when dealing with local problems. Henceforth, we do think a high-quality local dataset is needed and would also demonstrate the scientific interest of major reviews, such as *ESSD*, for a scientifically under-studied country.
- 2) When mentioning the lack of novelty in the approach, we do understand the criticisms of referee 2, as no "new" method is developed. However, we do think the novelty lies in the combination of known techniques and our unique pipeline/workflow. This study is fully reproducible from the sampling strategy to the final map produced. By combining the sampling strategy, campaign results, FTIR and laboratory measurements, FTIR model predictions, and DSM models, we propose a unique new approach inspired by Malone et al. (2022) but never applied in real conditions at a regional scale.

We do hope that these answers will incite referee 2 to reconsider the reasons for our application to *ESSD* journal.

**References used:**

- Ardizzone, F., Bucci, F., Cardinali, M., Fiorucci, F., Pisano, L., Santangelo, M., & Zumpano, V. (2023). Geomorphological landslide inventory map of the Daunia Apennines, southern Italy. *Earth System Science Data*, 15(2), 753–767. <a href="https://doi.org/10.5194/essd-15-753-2023">https://doi.org/10.5194/essd-15-753-2023</a>
- Błaszczyk, M., Luks, B., Pętlicki, M., Puczko, D., Ignatiuk, D., Laska, M., Jania, J., & Głowacki, P. (2024). High temporal resolution records of the velocity of Hansbreen, a tidewater glacier in Svalbard. *Earth System Science Data*, *16*(4), 1847–1860. https://doi.org/10.5194/essd-16-1847-2024
- Lorenz, C., Portele, T. C., Laux, P., & Kunstmann, H. (2021). Bias-corrected and spatially disaggregated seasonal forecasts: A long-term reference forecast product for the water sector in semi-arid regions. *Earth System Science Data*, *13*(6), 2701–2722. <a href="https://doi.org/10.5194/essd-13-2701-2021">https://doi.org/10.5194/essd-13-2701-2021</a>
- Malone, B., Stockmann, U., Glover, M., McLachlan, G., Engelhardt, S., & Tuomi, S. (2022). Digital soil survey and mapping underpinning inherent and dynamic soil attribute condition assessments. *Soil Security*, *6*, 100048. <a href="https://doi.org/10.1016/j.soisec.2022.100048">https://doi.org/10.1016/j.soisec.2022.100048</a>

**Citation**: https://doi.org/10.5194/essd-2025-418-RC2

---

## Author Comment (AC2)

*Referee 1 review*

Response to the recommender's assessment.

Changes in the text

essd-2025-418 "Soil information and soil property maps for the Kurdistan region, Dohuk governorate (Iraq)"

Bellat et al.

Review by D G Rossiter 29-Oct-2025

*Summary: This exceptionally-thorough and well-explained data paper presents details of the soils in the named region based on survey and models. It used modern methods (inference from MIR spectroscopy) as part of the soil properties determination. From this dataset a standard modern digital soil mapping (DSM) exercise was carried out to produce property maps over the study area. The maps were compared to the global SoilGrids v2.0 maps and, not at all surprisingly, had significantly better point evaluation metrics. All results and workflows are available under the FAIR concept. This paper can be a reference for how such a study can be carried out.*

Major Comments:

*1. I appreciate the thorough review of previous mapping efforts in the region, it is good to have these listed for reference. The brief review of major pedogenetic procesess is also appreciated. Similarly for the tectonic development, it places the study within context. The study's motivation is clear. Adherence to FAIR standards is appreciated. The entire workflow, all sources and products, are available, with DOI, and explained.*

Thank you so much for your positive feedback on our study. As you highlighted our aim was to have a fully see-thought workflow and process and data set accessible according to the FAIR principles.

*2. The Conclusions mainly repeat the Abstract and sections of the Discussion. I would appreciate a broader conclusion about the success of this study, the applicability of this kind of study to similar regions, the issues of global vs. local models, the main limitations to this kind of study, the importance of reproducibility and FAIR, etc. That is, after doing all this work, what do you conclude about the project?*

This comment is most welcome as I am sure the entire revision of the conclusion will help to improve the quality of the paper as suggest by reviewer #1.

l. "We developed a complete workflow for digital soil mapping at a regional scale in the Dohuk Directorate of the Kurdistan Region of Iraq, combining cLHS-driven sampling, MIR-based soil property prediction, and several machine-learning models to produce 50 soil property maps at 25 m resolution, as well as regional soil depth and soil class maps. Compared with *SoilGrids 2.0* and earlier local products, our models offer more locally relevant predictions and improved spatial detail, while also covering a broader set of soil properties and depth increments than previous regional studies. The soil class map further aligns with current WRB standards and benefits from greater observational density than earlier exploratory works.

Beyond these technical achievements, the study highlights the importance of integrating local measurements with models tailored to regional environmental gradients. Global products provide consistent baselines, but they cannot fully capture the geomorphological and topographic contrasts that drive soil variability at fine scales. The superior performance of our regional models demonstrates the complementarity between global and local approaches: global datasets remain essential for broad-scale comparisons, whereas locally calibrated workflows are crucial for operational land management, agricultural planning, and resource assessments.

The proposed workflow is fully transferable to other regions of similar size (~2,000 km²). Areas with comparable environmental conditions—such as western Iran, northern Syria, or parts of the Mediterranean basin—represent suitable candidates for direct methodological transposition. In addition, the time investment required for the entire process, from sampling design to final DSM production, is relatively modest: in our case, approximately one year (235 person-days). The datasets produced in this study also offer broader reusability, for example, as calibration material for MIR/FTIR spectral libraries (Safanelli et al., 2025; Viscarra Rossel et al., 2016) or as part of a regional or global soil profile archive database (Lachmuth et al., 2025).

A further contribution of this work is the provision of the first regional FAIR-compliant soil dataset (Crystal-Ornelas et al., 2022). Soil science research remains highly geographically imbalanced, with five countries producing more than 80% of global output (Cherubin et al, 2025). Southwestern Asia is sparsely represented, except for Iran, and no soil profiles from the Kurdistan Region of Iraq appear in the WoSIS database (Batjes et al., 2020) used by *SoilGrids 2.0*. (Poggio et al., 2021). Such data gaps reinforce global inequalities in environmental knowledge (Allik et al., 2020; Sonnenwald 2007) and limit the capacity of data-poor regions to benefit from international modelling initiatives. By openly sharing our dataset and workflow, we help reduce this imbalance and contribute to greater transparency and reproducibility in regional soil information systems.

Overall, this project demonstrates that high-resolution, locally informed digital soil mapping is both feasible and highly effective in data-poor regions. The workflow presented here substantially advances soil knowledge in Dohuk and provides a generalisable and reproducible model for improving soil information in other regions facing similar environmental and data constraints."

*3. Did you consider DSM for soil classes? Perhaps using a DSMART-like approach with your additional observations? This could be compared with Fig. 6. Obviously that is not to do in the paper, but was it considered and if so, why not attempted? Related to this, it is not clear how the soil class (not classification) map (Figure 6) was created. It's implied that this was expert judgement supplemented by observations, but it's not explicit. Also see comments below re: L443.*

You raised a very interesting point. Following the generalisation of DSM for quantitative soil information, it is likely that this will spread to qualitative information, such as soil classes. However, we had several reservations regarding automatic soil classes classification:

•       The first and maybe most prominent is the nature of soil classes. As described in the IUSS WRB (2022), soil classes can have multiple identities, with major and minor soil types present in the same landscape. Therefore, prediction models do not seem the most adapted tools – in our opinion – to discern the major and minor patterns of soil. However, it is true that with a random

forest model, you can assess the likelihood of a prediction and then select it manually based on your own expert opinion.

•	Second, the low influence of some physical variables on soil patterns in certain cases. Vegetation from Landsat or Sentinel bands can "trick" the model into predicting well-developed soil where there is just a low vegetation cover of Regosol, for example (in the foothills or the Little Khabur Valley). A good example is provided by Taghizadeh-Mehrjardi et al. (2014), who used a decision tree to predict soil classes. The predicted soil map (Figure 6) was poorly linked with topographic or geomorphological variables, which is uncommon for soil maps where these criteria are usually strongly linked (Graham 2006).

•	Third and last, our aim was not to conduct soil classes automatically from the beginning. The time spent in the field (3,5 months) was part of a general exploration of the region to understand the landscape as a whole (vegetation, hydrology, human installations, topography, geology, geomorphology). Therefore, we considered that an "expert-based knowledge" mapping is more suitable and reliable than an automatic one when time is allowed for the exploratory part of the survey.

The mention of DSMART-like is most welcome. We did look at the Easher et al. (2023) paper, and their results look promising. The question is whether this method is applicable in an arid or semi-arid context, whereas it is used here in a continental climate.

To clarify the use of "expert-based knowledge" mapping, we added a sentence.

l. "This resulted in a new regional soil map built on an expert-based knowledge method (Figure 6).".

References:

•	Easher, T. H., Saurette, D., Chappell, E., Lopez, F. de J. M., Gasser, M.-O., Gillespie, A., Heck, R. J., Heung, B., & Biswas, A. (2023). Sampling and classifier modification to DSMART for disaggregating soil polygon maps. *Geoderma*, *431*, 116360. https://doi.org/10.1016/j.geoderma.2023.116360

•	Graham, R. C. (2006). Factors of soil formation: Topography. In G. Certini & R. Scalenghe (Eds), *Soils: Basic Concepts and Future Challenges* (pp. 151–164). Cambridge University Press. https://doi.org/10.1017/CBO9780511535802.012

•	Taghizadeh-Mehrjardi, R., Sarmadian, F., Minasny, B., Triantafilis, J., & Omid, M. (2014). Digital Mapping of Soil Classes Using Decision Tree and Auxiliary Data in the Ardakan Region, Iran. Arid Land Research and Management, 28(2), 147–168. https://doi.org/10.1080/15324982.2013.828801

*4. Can you comment on the realism of patterns as seen in Figures 9--14? We have the point evaluation statistics, but the map shows a landscape. Do the elements we see there correspond to reality, of course by expert judgement? Are the fine details revealed by the 25 m resolution realistic or artefacts?*

The question of the realism of the patterns is closely linked with the soil map classes. Indeed, in a first place the presence of nudlithic Leptosol (l.155), so to say bare soil make any interpretation of

soil variables difficult to assess. Similar can be said based on the soil depth map where < 5 cm soil are observed (Figure 14). Although a filter could have been applied, we did not want to add this filter as the purpose of the paper was to provide a raw data set at a regional scale.

An example is the low presence of $C_t$, in all of the anticlines, and of $CaCO_3$ from the eastern mountainous are, at the lower depth increment of the soil (Figure 10 – 13), while most of the Bekhair anticline mountains are limestone. It is likely that these areas do have great concentration of $CaCO_3$ (Mustafa and Benni, 2014) but due to the lack of soil samples – and basically any *solum* – the model is unable to predict correctly the values.

For others as the MWD (50 – 70 cm) or EC (70 – 100 cm), the distribution pattern is directed by the model used. CART models are based on regression trees which make them very sensitive to any changes in the variables. Therefore, distribution of the variables does not entirely fit – to our opinion – the reality of the landscape at some point. The question remaining is weather you should keep a better model (in term of accuracy) but less reliable in explaining the variability of the landscape or a model with lower prediction results but potentially more fitted for landscape prediction (*e.g.* QRF)? Our choice was the one of the better model accuracies.

Regarding the other variables they do represents a reality of the landscape this is already discuss in Part 4.1.

Regarding the soil depth map (Figure 14) yes indeed it fits entirely the landscape observations and this is already discussed l.362 – 364.

One element also has to be taken into account it is the colour scale used in the figures 9 – 13. While they range from blue to red, the variables have in certain case very low SD or range (EC, MWD), which can make it more difficult to read. We did change the colour to the "classical" *viridis* palette.

References:

- *Mustafa, M. M., & Benni, T. J. (2014). Mineral ressources of the High Folded Zone. Iraqi Bulltin of Geology and Mining, Special Issue : 6, Article 6.*

*Detailed Comments:*

*The WRB 2006 has been replaced by WRB 2022: IUSS Working Group WRB: World Reference Base for Soil Resources. International soil classification system for naming soils and creating legends for soil maps, 4th ed., IUSS, Vienna, Austria, 234 pp., 2022. However, I think the definitions used in this paper have not changed.*

Thank you for this comment. Indeed, we did change the reference of the IUSS WRB 2006 into 2022 publication.

*L179-80 RUSLE: how were the parameters calibrated? Were they from one of the earlier (cited) studies? Especially the K value.*

Thank you for this comment. We added a sentence detailing the reference for each parameter and an annexe with the full equations used (appendix B). The process is also detailed in the supplementary material.

l. "In our RUSLE model (Appendix 2), we set *K* as the soil strength factor based on texture and organic carbon values (Kouli et al., 2009), and *C* was set with values from Morgan (2005) and

Swarnkar et al. (2018). Slope length and steepness factor *LS* were based on Desmet and Govers' method (1996), while the *R* factor from Morgan et al. (1984) was used. Finally, the conservation factor *P*, which could not be observed, has been set to 1 artificially, as suggested by Mehri et al. (2024). ”

*L227 "the different index" -> "the different indices"*

We modified this language, miswriting.

*L233 "We performed a standardisation of the predicted values of the texture on 100 % with TT.normalise.sum function (Moeys et al., 2024) and a additive-log ratio transformation (Aitchison, 1986) with the alr function (Tsagris et al., 2025)." This is not clear. Was the normalization following the MIR inference/wet lab measurements? And then were the alr variables used in the mapping, followed by back-transformation (as is done in SoilGrids v2.0)?*

Sorry if our explanation were not clear. You understood well, there was a small interference on the texture prediction based on the FTIR model. We did add an extra column to Table 2 to detail the measurements. These values were normalised on 100% before being transformed with the alr. The results of the best alr model were then transformed back to each texture variables. We added few sentences to clarify the whole process. We also specified that only continuous data were scaled when scaling the covariates. l. “resulting in 45 models and 50 maps in total (the three texture variables only include two alr models).”.

l. “We performed a standardisation of the predicted textures values from the Cubist model, with *TT.normalise.sum*”.

l. “Once the models were performed, the additive-log ratio was reversed into the three textures with the \texttt{alrInv} function, before being evaluated.”

 l. “We also scaled the continuous covariates [...]”.

*L235 "close to a normal distribution Liu et al. (2022)" -> "close to a normal distribution (Liu et al. ,2022)"*

We modified this part of the text.

*L236 "2021" refers to what?*

These data was referring to two references (Poggio et al., 2021; Varon-Ramirez et al., 2022). We added them.

*L258 "relative "simple"" -> "relatively simple"*

The modification was made.

*\S2.4.3 and throughout the paper: what is meant by "soil depth"? Is it the solum (zone of pedogenesis) or to bedrock/completely unweathered parent material?  This might be better termed "thickness" but "depth" is indeed commonly used. L365 "shallow and deep profiles" implies only the solum, is this correct?*

You highlighted a fundamental notion in soil science. We rely on the IUSS WRB 2022 definition of the *solum* concentrated in the pedogenesis process layers. With “soil depth,” we imply all horizons of the pedogenesis process (*solum*) down to the top of the mineral layer/weathered consolidated

rock (C horizon), as detailed on p. 225 of IUSS WRB 2022. To clarify this point, we added a sentence in the article.

l. "By soil depth, we defined the height of the active horizons in the pedogenesis process (IUSS WRB 2022), the so-called *solum*. The measurement is taken from the top of the *solum* downward to the mineral layer (C horizon) or consolidated rock (R horizon).".

*L295 the correct reference for PICP is Eq. (2) of Malone (2011) not 2017. The formula is not found in Malone 2017. Malone, B. P., McBratney, A. B., and Minasny, B.: Empirical estimates of uncertainty for mapping continuous depth functions of soil attributes, Geoderma, 160, 614–626, https://doi.org/10.1016/j.geoderma.2010.11.013, 2011. This equation and the others need definitions of the symbols, although some are standard. For PICP what is "v"? I learn from Malone 2011 it is "he number of observations in the validation [better, evaluation] dataset". What is "PL"? L, U as lower, upper limits can be inferred. Finally, the description "we used the prediction interval coverage probability to evaluate the corresponding prediction within an interval" is not clear. The Malone 2011 description is, to me, clearer: "the PICP is the probability that all observed values fit within their prediction limits".*

We would like to thank referee 1 for pointing out the original publication of the PICP in the DSM application. We added the reference to Malone et al. (2011) and their formula, while also retaining the original publication of the PICP from Shreshta et al. (2006).

We did change the sentence explaining the PICP as described below.

l. "Finally, for the QRF models, we used the prediction interval coverage probability (PICP), which correspond to the probability that all observed values fall within their prediction intervals, here set at 90 % "

More generally, we added as you suggested a description of all equation formula excepted for the RPIQ.

*L309 It's interesting that silt is so poorly predicted, yet most of these soils are on the silty edge of the texture triangle. And, clay and sand are in Category A and B. Can you explain why the poor result for silt, even though there is a lot of it and with a good range in these samples? This is mentioned on L387.*

Thank you for this reflection on the results and their interpretation. Explaining the high silt content in the samples is difficult without a more detailed geomorphological and granulometric investigation. However, some leads can already be found. First, the relatively low sand content is due to numerous erosional agents in the region (water action, wind action, weathering, slope processes), but the relatively low effect of wind action compared to the southern and western parts, where Arenosol can be found (Buringh 1957). The low clay content is due to the Simele plain's higher elevation relative to the Tigris (a 30–50 m difference). These values are consistent with what Abdulrahman et al. (2020) observed in their paper in the Simele Plain.

Predicting silt is difficult for mid-infrared detectors. Hobeley and Prater (2019) highlighted this issue of silt prediction by stating, "This resulted in the best prediction of the silt fraction, which is traditionally poorly predicted by vis–NIR". These poor results are also reported by Viscarra Rossel and Webster (2012), Janik et al. (2016), and Ng et al. (2022). In the case of Lacerda et al. (2016), the results of the silt prediction are even regarded as so low that they choose to only calculate the silt based on the equation [100 – clay% - sand%]. This approach was not used because it does not allow

an estimate of the silt error and does not account for the possibility that the predicted texture amount is lower or higher than 100% as in a full texture prediction. However, it seems that vis-NIR yields better results than MIR (Ng et al., 2022), which may also explain our poor predictive performance. Another lead is discussed by Hobeley and Prater (2019), stating that RF would perform better than Cubist or other models for silt prediction. We do not fully agree on this statement, as if indeed it yields better results, the silt stays in their publication, the "most badly predicted" variable with an RPD of **3.86** (Table 2). We here preferred consistency with a unified model (Cubist) for all parameters.

As for the chemical or physical reason for such low prediction of silt, it can reside in several reasons that would have to be explored:

- The higher water concentration in clay helps the spectral abortion.

- The quartz mineral is highly present in sands, which have a clear signature for spectral detection (Lacerda, 2016).

- The physical structure of the silt molecule (Janik et al. 2016).

We added the following sentence to detail these potential effects.

l. "Regarding the silt prediction, several reasons could explain its relatively low reliability. First, the use of MIR instead of vis-NIR, and the use of the Cubist model instead of RF, both of which yield better results (Hobeley and Prater, 2016). Second, the nature of the silt molecule, which is more challenging to detect than others (Janik, 2016; Lacerda, 2016)."

References:

- Hobley, E. U., & Prater, I. (2019). Estimating soil texture from vis-NIR spectra. *European Journal of Soil Science, 70*(1), 83–95. https://doi.org/10.1111/ejss.12733

- Janik, L. J., Soriano-Disla, J. M., Forrester, S. T., & McLaughlin, M. J. (2016). Effects of soil composition and preparation on the prediction of particle size distribution using mid-infrared spectroscopy and partial least-squares regression. *Soil Research, 54*(8), 889–904. https://doi.org/10.1071/SR16011

- Lacerda, M. P. C., Demattê, J. A. M., Sato, M. V., Fongaro, C. T., Gallo, B. C., & Souza, A. B. (2016). Tropical Texture Determination by Proximal Sensing Using a Regional Spectral Library and Its Relationship with Soil Classification. Remote Sensing, 8(9), 701. https://doi.org/10.3390/rs8090701

- Ng, W., Minasny, B., Jeon, S. H., & McBratney, A. (2022). Mid-infrared spectroscopy for accurate measurement of an extensive set of soil properties for assessing soil functions. Soil Security, 6, 100043. https://doi.org/10.1016/j.soisec.2022.100043

- Rossel, R. A. V., & Webster, R. (2012). Predicting soil properties from the Australian soil visible–near infrared spectroscopic database. *European Journal of Soil Science, 63*(6), 848–860. https://doi.org/10.1111/j.1365-2389.2012.01495.x

L346 "river bakns" -> "river banks".  Spell-check.

Thank you for spotting this error, the change was made.

*\S4.3 Another interesting comparison with SG2 would be the prediction ranges. SG2 likely smooths more than this study, see Table 7 where the Q1-Q3 range is always much narrower. This can be brought out in the text -- the interesting discussion is about global vs. local models. The SG2 maps are much more uniform than the maps from this study.*

You raised an interesting point. Due to its coarser size and also the wider scale of measurement observed the *SoilGrid.2.0* product is likely to produce narrower. We did provide an additional table of the metrics from *SoilGrid 2.0.* (Appendix F) which we added to this discussion.

l. "The standard deviation of the *SoilGrids 2.0* product is smaller, except for the pH, than for our prediction models (Table 7). This shows a narrower distribution of values, likely due to the wide range of input data used for the *SoilGrids 2.0*. The diversity of soil types and input data at the global scale makes the *SoilGrids 2.0* model respond relatively homogeneously at the regional scale."

l. "We also compared the evaluation metrics of our predicted values with those obtained from *SoilGrids 2.0* (Appendix ). Overall, our models outperformed *SoilGrids 2.0* across all evaluation metrics for pH, $N_t$, Corg, sand, silt, and clay at all depth intervals. The only exception was the sand $R^2$ score at the 60–100 cm depth interval, which was slightly higher for the *SoilGrids 2.0* model."

*L387 "should be interpreted with caution—consistent"... with what?*

Indeed, an extra word was added for not reason. We removed "—consistent"

*L392 "Abdulrahman et al. 2020 " -> "Abdulrahman et al. (2020)".  L390 maybe make it explicit here that this is not a DSM product, rather an expert updating from field work and manual interpretation of remote sensing products (correct?).*

Yes, the work of Abdulrahman et al. (2020) consisted of a punctual soil profile sampling study. We used their observations as a comparison to our soil profiles results and to also state that soil profiles (*solum*, see \S2.4.3) with a depth greater than 100 cm are concerning few soil profile (less than the half) and even some have to be treated with caution (B2ss and B2hk). calcrete layer can be very similar with C horizons. We rephrase the sentence of l.392

l. "[…] although other local profile observations (Abdulrahman et al. 2020) tend to show that profiles below 100 cm are uncommon"

*L401 the Hazelton & Murphy guidelines are for conventional mapping, not DSM. They are expressed in terms of map scale and cm^2 of printed map. Here the product is digital at 25 m resolution. How is the density here converted to match these guidelines? The argument about cLHS is much more relevant for DSM using machine learning from covariates.*

You are entirely right. This comparison with Hazelton & Murphy mapping convention is relevent for DSM products. We removed the sentence mentioning it.

*L434 formatting problem with the URL https://mathias-bellat.shinyapps.io/Northern-Kurdistan-map, which goes over a line break so gives a 404 error if not manually adjusted*

Thank you for noticing this error we modified the text and implementation in LaTEX.

*L443 "shallower resolution " -> "higher resolution"? And what is that resolution? It's nowhere stated. L435 says 1:200 000 scale, which implies polygons with a minimum legible delineation*

*(MLD) of 160 ha (0.4 cm^2 on the map). But L390 "The updated soil classification map (Figure 6) must be interpreted with care, specially at micro-scale (<1:50,000)..." implying a smaller MLD.*

You are right, the observations of Buringh (1960) from the previous soil map (1957) are made at the national scale and no information were given for its sampling or whatever. The only comment on the map is made p. 291 (Buringh 1960) "The topographical map of Iraq, quarter of an inch to 1 mile; The provisional geological map of Iraq, 1:2,000,000, 1937; The soil conservation map of Iraq, 1:2,000,000 (Gibbs, 1953); The forest map of North-eastern Iraq, 1:850,000; Some aerial photo-mosaics; Some soil and land classification maps of project areas.". From the reading of the book it is likely that few profiles < 20 where done the north-western Kurdistan region. The only information is the scale of the map which was at 1:1 000 000.

Our own resolution indeed was not stated but based on the number of observations l.399 (53.5 per 1,000 km2) which regarding Hazelton and Murphy (2016) table 1.2 would be at a reconnaissance 2 : 250 000 scale. However, the recent development of G.I.S and remote sensing has made the relevance of map scale less import. In the supplementary files the complete vector files are available making possible mapping at any scale.

However, to reduce the ambiguity we change the sentence.

l. "[…] must be interpreted with care, especially for local agricultural or construction planning."

References:

- Buringh, P. (1957). Exploratory Soil Map of Iraq (Division of soils and agricultural chemistry, Directorate General of Agricultural Research and Projects,) [Soil map]. Ministry of Agriculture. https://esdac.jrc.ec.europa.eu/content/exploratory-soil-map-iraq-map-1

- Buringh, P. (1960). Soils And Soil Conditions In Iraq. Minstry of Agriculture (Iraq). http://archive.org/details/buringh-p.-soils-and-soil-conditions-in-iraq.-republic-of-iraq

- Hazelton, P. A., & Murphy, B. W. (2016). Interpreting soil test results: What do all the numbers mean? (Third edition). CSIRO Publishing.

*Figure 6 suggest that this is a polygon map.*

Sorry, we did change the map into a raster figure instead of a vector figure. The figure was also slightly updated to include Türkiye soils. We also changed the others large scale maps and figure 2 which were vectorised maps.

*Figure 3 the inset showing the region is not needed, that has already been shown in Figure 1 and can be found from the coordinates on the main map.*

Indeed, the additional small map on the Figure 3 was repetitive regarding the presence of Figure 1, we did remove it.

*Figure 8 bicolor key y-axis partially obscured*

We apologise, reviewer #1, but we did not understand this comment. In our pre-print version, the y-axis was not obscured. However, we did change the legend position.

**Citation**: https://doi.org/10.5194/essd-2025-418-RC1

---

## Author Comment (AC3)

*Recommender 3 review*

Answer to the recommender review

Changes in the text

essd-2025-418 "Soil information and soil property maps for the Kurdistan region, Dohuk governorate (Iraq)"

Bellat et al.

Review by Bas Kempen, 23 Jan 2026

*A comprehensive paper in a critically under represented geographical area when it comes to soil profiles/ digital soil mapping. Care has been taken with the landscape characterisations including tectonic development and parent material climate and vegetation and geomorphology and soils, as well as maps and photographs to allow the reader to really understand the study area. While the methods are not necessarily 'new' themselves, it is an important application of state of the art methods, the novel part of this study is the study area. The output maps are compared to SoilGrids, a global model, with inputs from WoSIS, the study mentions that WoSIS has low sampling density in this area, highlighting the need for such studies, in addition to the scarcity of other options in the area.*

*I agree with reviewer 1 that this is a well written and thorough study. This study is a good example of adhering to FAIR and open metadata standards, for not only the data but the methodology, and that is commendable. The methods are comprehensively described, and fit their purpose well. All methodological aspects including the code are not only made available following FAIR principles, but thoroughly documented and explained at https://mathias-bellat.github.io/DSM-Kurdistan/digital-soil-mapping.html#visualisation-and-comparison-with-soilgrid-product, creating a fine example of a fully reproducible study and the input and output data themselves are a much needed addition to more or less non-existent openly available soil data in the area. With this, this manuscript fits well within the scope of ESD.*

*One of the reviewers mentioned the limited geographic scope of the paper. I do agree that this scope is limited but having a DSM study published for a region like Kurdistan (Iraq) is worthwhile and to me an a welcome addition to the body of literature on this topic. Especially given that the authors make their results as well as data open from which other DSM efforts (e.g. SoilGrids) can profit. This is much appreciated.*

*Having said this, I do have a few comments and questions regarding the manuscript, particularly concerning the methodologies, which in my view require some further explanation and clarification. I encourage the authors to address these points, after which I would recommend publication of this article in ESSD.*

Thanks you for this feedbacks on our study. We are please that our aims, for producing a fully reproducible and open methodology in an under-represented areas, has been acknowledged and highlighted by the recommender. Indeed, we do hope that this study and in particular the results would be used in improving global databases, such has WoSIS, or models as *SoilGrids.*

*Main comments*

*Lines 230-231: Could the authors explain the decision to model each depth layer separately instead of developing one model per property with using the depth as an explanatory variable? Modelling each depth separately is a valid approach, but I would like to understand the reason why the authors took this approach.*

Indeed, this choice is motivated by two main arguments:

- First is the question of methodological reproducibility. We based our approach on similar studies, such as the *SoilGrid.2* (Poggio et al., 2021), national soil texture modelling (Varón-Ramírez et al., 2022) – a very well-performed model which inspired part of our methodology - and all these models and maps are using independent depth increment models. Our methodology was also inspired by Malone et al. (2022), who produced independent depth models based on different soil samplings.

- Second is the question of whether or not the soil depth should be used as a predictor. Besides the covariance/collinearity, which would be really high due to the numerous covariates used in both modelling, some scholars have shown that a 3D data-driven approach should be taken with caution (see Ma et al., 2021).

To continue this discussion, we considered setting a "mask" for soil depths below 5 cm. However, the purpose of this study, and of ESSD to some extent, is provide more **raw** data that can be interpreted by any researcher regarding their own discipline. A geochemist or people working with remote sensing might find some interest in having, even if low, a partial soil covering over the whole study area. While researchers working on pedogenesis of deeper soils will likely exclude all soils with depths lower than 5–15 cm.

*Lines 246-247: I do not understand the rationale for combining data splitting (80/20) with (repeated) cross validation. Cross-validation already produces an independent prediction for each data point, from which accuracy metrics can be calculated. What was the motivation for embedding CV within a data-splitting framework? And how does this then work? If CV was performed on the 80% training subset, this would give CV-predictions only for these points. I wonder how predictions for the 20% test set were obtained. Which trained model was used to predict at the points in the test set? In case of normal data splitting this would be the model trained on the training dataset. However, by running CV on the training dataset there is no single trained model but multiple (here 10) fold-specific models.*

Thank you for this comment. You raise an important point on the preparation and training of models. We do think there is often a misunderstanding between **validation** and **test** data. The validation set serves as an independent set during training to tune the model's parameters. The test values serve only as a "control" of the model based on the evaluation process at the end. This set is then fully independent of any prior training and better reflects the model's adaptability and **interoperability.** The introductory book by Alpaydin provides a well-explained analogy of the **train/validation/test sets** (Alpaydin, 2014, p. 40).

Regarding the selection from this CV, we selected the best model and evaluated it on the test set, reporting the **test evaluation metrics** in the table. We used the `train` function from *caret* and the basic `predict` function from stats R to predict the test set as :

*"predict(Evaluation$Models[[i]][[j]], X_test)"*

The final will be the training being the `Evaluation$Models[[i]][[j]]$finalModel` which is a model based on the training with the best hyperparameters collected from the 10 folds of each 3 repetitions ending in a total of **30** models trained with the tuning grid (depending on each type of model used). For example, for the **Cubist** model, we used the following tuning grid *"expand.grid(committees = c(1, 5, 10, 15, 20), neighbors = c(0, 1, 2, 3, 5, 7, 9))"*. In total, for one **Cubist** training on a variable, we have 35 (hyperparameters) x 30 (resampling) = **1050** independent models.

The `train` function then reselects the best combination as explained in the ***Caret*** manual: *"The combination with the optimal resampling statistic is chosen as the final model and the entire training set is used to fit a final model."*.

We did add a sentence to detail the split strategy with clarification on **validation/test** sets:

"For each soil depth, we used a two-step evaluation. First, we set aside 20% of the data as an independent test and did not use it for any model decisions. Then, we used the remaining 80% as the training set and ran repeated k-fold cross-validation to tune the model and choose the final settings. After the final settings were chosen, we trained the final chosen model on the full 80% training set and used this single model to make predictions for the held-out 20% test set."

*In addition, the manuscript states that CV was repeated three times? While repetition may improve robustness when data are limited or when using a small number of folds, with 10-fold CV I would expect only minimal variation between the repeats?*

The point of repeating the CV is not to improve the model but to reduce bias sampling. As our training sample sizes are quite small (98, 90, 86, 87, and 74, respectively, due to the test set split), and we are handling spatial data that is often prone to overfitting (Brus, 2022), we need to reduce the bias introduced by our sampling strategy.

Furthermore, the RMSE values varied slightly across repetitions, suggesting that a single k-fold CV could have yielded a biased estimate, especially given the spatial nature of the data. The repeated CV provides a more stable and reliable assessment. A recent study by Lumumba et al. (2024) found lower variance and greater stability of cross-CV compared with standard CV.

We therefore assessed each model's variability to determine whether any fold showed massive overfitting. **None was witnessed (< 10% variance in the RMSE)**, and an extra line of code was added to a new table for each depth, "Repetition_variance.txt," in the supplementary files.

We did add an extra sentence to clarify our use of the three-time repetition CV:

"This resampling strategy allows us to avoid potential overfitting due to the small size of our training data set (< 100) and the spatial nature of our data."

*Overall, the validation approach seems a bit overcomplicated. The authors may well have had sound reasons for adopting this approach, but in case the rationale and precise implementation need to be explained more clearly.*

We already answered part of the question in the above comments. However, to complete our answer, we do not think the resampling process chosen is inadequate. We are facing, as many DSM studies have, two main issues:

- The bias in our samples due to their spatial nature, inducing either a high number of repetitions - our chosen approach – or a spatial cross-validation such as the *BlockCV* package (Valavi et al., 2019) or another type of spatial sampling (Schwarz et al., 2019).

- The size of our data is rather limited, less than 100 samples for the training phase. The number of repetitions has to be consistent to reduce variance and limit overfitting. An alternative approach would have been to use leave-one-out (LOOCV) sampling.

To address both issues, we adopted a 3-fold repeated 10-fold CV. As it might be more time-consuming than a "classical" CV (Lumumba et al. 2024), it yields less variance than a simple CV and is less demanding in terms of computer resources than a spatial LOOCV.

Overall, if the three repetitions do not improve the accuracy or sensitivity of the models, they do help reduce the potential bias from overfitting.

*Lines 268-273: The description of the ensemble modelling approach is unclear to me and would benefit from additional detail. Specifically, it is not clear which 'conditions' (l. 269) are being referred to, what criteria were used to select 'the best one', and how the individual model predictions were combined in the ensemble? While relevant literature is cited, I believe that a few additional lines better outlining the implementation of the ensemble approach would improve clarity to the reader.*

Sorry for the unclearness of our sentence. You are right, the term "conditions" is totally misused here. We used a stacked regression. We did clarify the meaning of the ensemble model and the "meta-model" used here.

"Finally, an ensemble model is created by stacking the five previously trained models. A random forest serves as the "meta-learner", which takes the predictions of the base models as input and learns how to optimally combine them using the caretStack function."

*Other comments*

*Line 10: The summation signs should be removed I believe.*

We did remove the signs as in line 14 of the abstract.

*Line 10-11: Reference is made to 'local models' (compared to the regional models the authors developed and the SoilGrids global model). It is unclear to me where the 'local models' refer to and what the basis is for the claim of the authors.*

Thank you for pointing out this mistake. We wanted to refer to 'regional' models, which are intended for use at finer scales (e.g. regions, cities or towns) than national or worldwide models. We did change the term "local" to "regional". We do not differentiate between our models and the ones from Yousif et al. (2023) in term of scale.

*Line 14: I believe the minus sign in the superscript should be removed, assuming the RMSE values are reported for the transformed depth data. After applying the square-root transformation, the depth unit becomes cm^0.5. The unit of the MSE metric would then be cm, and taking the square root to obtain the RMSE would again give value in cm^0.5. Table 5 also reports the RMSE unit in cm^0.5. I assume the unit of the MAE (Table 5, l. 337) is also cm^0.5?*

Thank you for noting this mistake. However, we recomputed the soil depth with a fully reproducible workflow (adding python GEE API), during the update we changed the code to produce directly transformed metrics. We therefore, removed all the reference to transformation ($cm^{0.5}$) in the text.

*Line 148: Reference is made to WRB 2006. Can the authors confirm if this was also given that there are more recent versions of the WRB? The latest from 2022 I believe.*

Comments from recommender 1 were pointing out to this mistake. We did change all the references according to WRB 2022, which did not yield major difference for aridic and semi-aridic soils.

*Line 178: "potential soil layer" - inconsistent naming – previous paragraph is "potential soil properties"*

Thank you for point out this mistake, a change was done.

*Line 185: What are the 'layers' here? Are these the soil horizons or are these the layers for which samples were collected (l. 186)? Please clarify.*

You are entirely right this induced to a misunderstood. We do mean "horizons" in here.

We changed for "soil horizons' depths"

*Line 187: How is 'topsoil' defined here? Is this the 0-10 layer? Explain a bit more how the ring samples were taken. E.g. where was the ring sample was taken in the topsoil layer: from the top, in the middle of the layer …?*

Sorry for the lack of clarity in this element. We sampled the top of the soil after removing all vegetation and loose material. Related to the height of the ring (c. 5 – 7 cm), only the upper part of the top soil, 0 – 10 cm, was then sampled.

We rephrase the sentence to clarify:

"After removing the surface litter and loose sand, the sampling ring was used on the 0 –10 cm soil layer."

*Lines 236-237: References seem to be incomplete. Only years are mentioned (2021,2022)*

Sorry for this error the new version do not have this typo. The related references are (Poggio et al., 2021; Varon-Ramirez et al., 2022).

*Line 243: Could expand on why RFE was not restrictive enough/ restrictive enough for what?*

The results of the RFE did provide optimal tuning for a larger number of covariates (*c.* > 20) compared to the relatively low number for the Boruta (< 10). Therefore, the computing time needed for modelling would have been greater for limited results.

Details of the RFE results can be found in the supplementary material (7 – DSM/export/RFE), including a detailed graph of the optimal RFE selection and the exported variables selected by RFE.

*Line 372: Why were SoilGrids extremes removed?*

This question is most welcome. The *SoilGrids* relies on land use imagery for masking some areas (rivers, lakes, oceans or cities). Due to the coarse resolution of the models (250 m) these areas and their neighbouring pixels are often highly outliers. To overcome this effect two option were possible, either using the same mask (which is not provided) or smoothing these outliers. We choose the second

option mainly for practical issues and as the number of pixel presenting outliers were not highly consistent (*c. < 5%*).

*Line 401: Limitations are addressed in the paper – I am not sure if the point density, even though notably higher than WoSIS, still warrants mapping at 25m, the reference to Hazelton and Murphy (referencing cartographic scales) seems a bit of a jump. Instead of referencing Hazelton and Murphy I would rather compare to other regional DSM studies.*

You are entirely right this was also an underlying comment for the recommender 1. This comparison with "traditional mapping" density from Hazelton and Murphy (2016) is not relevant. We removed the entire sentence.

*The Conclusion section reads like an abstract. Reviewer 1 already commented on this and based on that, the cauthors revised the text and I believe with that revision this comment is addressed sufficiently.*

Thank you for having reading the revised version of the conclusion. Indeed we changed in depth this conclusion regarding recommender 1 comments. We provide here the text if ever it is not easily readable on the comment version from ESSD.

l. "We developed a complete workflow for digital soil mapping at a regional scale in the Dohuk Directorate of the Kurdistan Region of Iraq, combining cLHS-driven sampling, MIR-based soil property prediction, and several machine-learning models to produce 50 soil property maps at 25 m resolution, as well as regional soil depth and soil class maps. Compared with SoilGrids 2.0 and earlier local products, our models offer more locally relevant predictions and improved spatial detail, while also covering a broader set of soil properties and depth increments than previous regional studies. The soil class map further aligns with current WRB standards and benefits from greater observational density than earlier exploratory works.

Beyond these technical achievements, the study highlights the importance of integrating local measurements with models tailored to regional environmental gradients. Global products provide consistent baselines, but they cannot fully capture the geomorphological and topographic contrasts that drive soil variability at fine scales. The superior performance of our regional models demonstrates the complementarity between global and local approaches: global datasets remain essential for broad-scale comparisons, whereas locally calibrated workflows are crucial for operational land management, agricultural planning, and resource assessments.

The proposed workflow is fully transferable to other regions of similar size (~2,000 km²). Areas with comparable environmental conditions—such as western Iran, northern Syria, or parts of the Mediterranean basin—represent suitable candidates for direct methodological transposition. In addition, the time investment required for the entire process, from sampling design to final DSM production, is relatively modest: in our case, approximately one year (235 person-days). The datasets produced in this study also offer broader reusability, for example, as calibration material for MIR/FTIR spectral libraries (Safanelli et al., 2025; Viscarra Rossel et al., 2016) or as part of a regional or global soil profile archive database (Lachmuth et al., 2025).

A further contribution of this work is the provision of the first regional FAIR-compliant soil dataset (Crystal-Ornelas et al., 2022). Soil science research remains highly geographically imbalanced, with five countries producing more than 80% of global output (Cherubin et al, 2025). Southwestern Asia is sparsely represented, except for Iran, and no soil profiles from the Kurdistan Region of Iraq appear in the WoSIS database (Batjes et al., 2020) used by SoilGrids 2.0. (Poggio et al., 2021). Such data gaps reinforce global inequalities in environmental knowledge (Allik et al., 2020; Sonnenwald 2007) and limit the capacity of data-poor regions to benefit from international modelling initiatives. By openly sharing our dataset and workflow, we help reduce this imbalance and contribute to greater transparency and reproducibility in regional soil information systems.

Overall, this project demonstrates that high-resolution, locally informed digital soil mapping is both feasible and highly effective in data-poor regions. The workflow presented here substantially advances soil knowledge in Dohuk and provides a generalisable and reproducible model for improving soil information in other regions facing similar environmental and data constraints."

*Figure 2: What is a negative site? – soil samples*

Sorry for not explaining this element. It refers to the line 275 – 276: "The soil depth was measured from 0 to 100 cm on the 122 sampling sites; we added 25 zero values from remote sensing imagery observation on bare rock points.". We did added the mention of **negative site**

" sampling sites; we added 25 zero values (negative sites) from"

*Technical corrections*

- *Line 21: influence local ecosystems -> influence on local ecosystems*
- *Line 28: includes -> include*
- *Line 33: gives information on its ability to fit or not for agricultural purposes, but also to better understand -> gives information on its ability to fit agricultural purposes and helps to better understand*
- *Line 35: Governate -> governate (not capitalised anywhere else in the paper)*
- *Line 70: cluster -> conditional*
- *Line 75 & 265: Hengl and Robert -> Hengl and MacMillan*
- *Line 74: a raw -> one raw*
- *Line 129: climax -> climate*
- *Line 255: McBradney -> McBratney*
- *Line 229: remove 'part'*
- *Line 234: a additive -> an additive*
- *Line 247: state of the art the art -> state of the art models*
- *Line 270: approached -> approach*
- *Line 346: bakns -> banks*

- *Line 388: remove 'consistent'*

- *Line 401: Hazelton and Murphy pg5 -> pg4*

- *Line 410: LU/C -> land use/cover (it is not used previously)*

- *Line 425: profiles depth measurement -> profile depth measurements*

- *Line 441: world -> global*

- *Line 443: shallower resolution -> higher resolution*

- *Table 3: Modis brightness index in the wrong column*

- *Appendix B: units column could be named differently. It contains a mixture of units, formulas, ranges.*

We deeply thank the recommender for all of these comment. We did make all the changes needed. As for the Appendix B we did change the name of the column to **"measure".**

**References:**

Alpaydin, E. (2014). Introduction to machine learning (Third edition). The MIT Press.

Brus, D. J. (2022). *Spatial Sampling with R* (1re éd.). Chapman and Hall/CRC. https://doi.org/10.1201/9781003258940

Ma, Y., Minasny, B., McBratney, A., Poggio, L., & Fajardo, M. (2021). Predicting soil properties in 3D : Should depth be a covariate? Geoderma, 383, 114794. https://doi.org/10.1016/j.geoderma.2020.114794

Malone, B., Stockmann, U., Glover, M., McLachlan, G., Engelhardt, S., & Tuomi, S. (2022). Digital soil survey and mapping underpinning inherent and dynamic soil attribute condition assessments. Soil Security, 6, 100048. https://doi.org/10.1016/j.soisec.2022.100048

Poggio, L., de Sousa, L. M., Batjes, N. H., Heuvelink, G. B. M., Kempen, B., Ribeiro, E., & Rossiter, D. (2021). SoilGrids 2.0 : Producing soil information for the globe with quantified spatial uncertainty. *SOIL*, *7*(1), 217-240. https://doi.org/10.5194/soil-7-217-2021

Schratz, P., Muenchow, J., Iturritxa, E., Richter, J., & Brenning, A. (2019). Hyperparameter tuning and performance assessment of statistical and machine-learning algorithms using spatial data. Ecological Modelling, 406, 109-120. https://doi.org/10.1016/j.ecolmodel.2019.06.002

Varón-Ramírez, V. M., Araujo-Carrillo, G. A., & Guevara Santamaría, M. A. (2022). Colombian soil texture : Building a spatial ensemble model. *Earth System Science Data*, *14*(10), 4719-4741. https://doi.org/10.5194/essd-14-4719-2022

Yousif, B. S., Mustafa, Y. T., & Fayyadh, M. A. (2023). Digital mapping of soil-texture classes in Batifa, Kurdistan Region of Iraq, using machine-learning models. Earth Science Informatics, 16(2), 1687-1700. https://doi.org/10.1007/s12145-023-01005-8

**Citation**: https://doi.org/10.5194/essd-2025-418-RC3